# Catastrophic Fisher Explosion: Early Phase Fisher Matrix Impacts Generalization

## Abstract

The early phase of training has been shown to be important in two ways for deep neural networks. First, the degree of regularization in this phase significantly impacts the final generalization. Second, it is accompanied by a rapid change in the local loss curvature influenced by regularization choices. Connecting these two findings, we show that stochastic gradient descent (SGD) implicitly penalizes the trace of the Fisher Information Matrix (FIM) from the beginning of training. We argue it is an implicit regularizer in SGD by showing that explicitly penalizing the trace of the FIM can significantly improve generalization. We further show that the early value of the trace of the FIM correlates strongly with the final generalization. We highlight that in the absence of implicit or explicit regularization, the trace of the FIM can increase to a large value early in training, to which we refer as catastrophic Fisher explosion. Finally, to gain insight into the regularization effect of penalizing the trace of the FIM, we show that it limits memorization by reducing the learning speed of examples with noisy labels more than that of the clean examples, and 2) trajectories with a low initial trace of the FIM end in flat minima, which are commonly associated with good generalization.

## 1 Introduction

Implicit regularization in gradient-based training of deep neural networks (DNNs) remains relatively poorly understood, despite being considered a critical component in their empirical success (Neyshabur, 2017; Zhang et al., 2016; Jiang et al., 2020b). Recent work suggests that the early phase of training of DNNs might hold the key to understanding these implicit regularization effects. Golatkar et al. (2019); Keskar et al. (2017); Sagun et al. (2018); Achille et al. (2019) show that by introducing regularization later, a drop in performance due to lack of regularization in this phase is hard to recover from, while on the other hand, removing regularization after the early phase has a relatively small effect on the final performance.

Other works show that the early phase of training also has a dramatic effect on the trajectory in terms of properties such as the local curvature of the loss surface or the gradient norm (Jastrzebski et al., 2020; Frankle et al., 2020). In particular, Achille et al. (2019); Jastrzębski et al. (2019); Golatkar et al. (2019); Lewkowycz et al. (2020); Leclerc & Madry (2020) independently suggest that rapid changes in the local curvature of the loss surface in the early phase critically affects the final generalization. Closely related to our work, Lewkowycz et al. (2020); Jastrzębski et al. (2019) show that using a large learning rate has a dramatic effect on the early optimization trajectory in terms of the loss curvature. These observations lead to a question: what is the mechanism by which regularization in the early phase impacts the optimization trajectory and generalization? We investigate this question mainly through the lens of the Fisher Information Matrix (FIM), a matrix that can be seen as approximating the local curvature of the loss surface in DNNs (Martens, 2020; Thomas et al., 2020).

Our main contribution is to show that the implicit regularization effect of using a large learning rate or a small batch size can be modeled as an implicit penalization of the trace of the FIM ($\mathrm{Tr}(\mathbf{F})$) from the very beginning of training. We demonstrate on image classification tasks that the value of $\mathrm{Tr}(\mathbf{F})$ early in training correlates with the final generalization performance across settings with different learning rates or batch sizes. We then show evidence that explicitly regularizing $\mathrm{Tr}(\mathbf{F})$ (which we call Fisher penalty) significantly improves generalization in training with a sub-optimal learning rate. On the other hand, growth of $\mathrm{Tr}(\mathbf{F})$ early in training, which may occur in practice when using

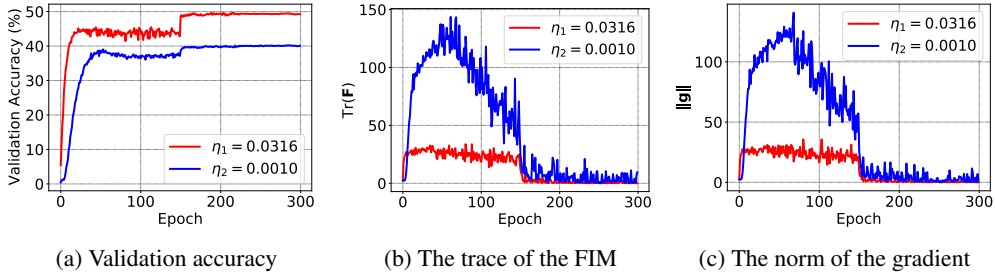

Figure 1: The catastrophic Fisher explosion phenomenon demonstrated for Wide ResNet trained using stochastic gradient descent on the TinyImageNet dataset. Training is done with either a learning rate optimized using grid search ($\eta_1 = 0.0316$, red), or a small learning rate ($\eta_2 = 0.001$, blue). Training with $\eta_2$ leads to large overfitting (left) and a sharp increase in the trace of the Fisher Information Matrix (FIM, middle). The trace of the FIM is closely related to the gradient norm (right).

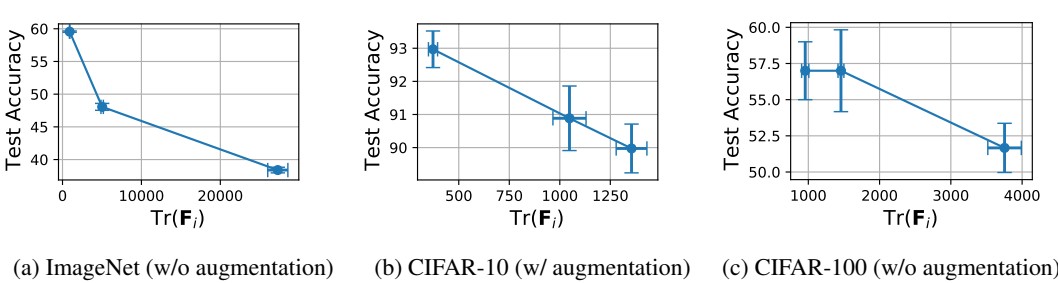

Figure 2: Association between the value of $\mathrm{Tr}(\mathbf{F})$ in the initial phase of training ($\mathrm{Tr}(\mathbf{F_i})$) and test accuracy on ImageNet, CIFAR-10 and CIFAR-100 datasets. Each point corresponds to multiple seeds and a specific value of learning rate. $\mathrm{Tr}(\mathbf{F_i})$ is recorded during the early phase of training (2-7 epochs, see the main text for details). The plots show that early $\mathrm{Tr}(\mathbf{F})$ is predictive of final generalization. Analogous results illustrating the influence of batch size are shown in Appendix A.1
.

a relatively small learning rate, coincides with poor generalization. We call this phenomenon the catastrophic Fisher explosion. Figure 1 illustrates this effect on the TinyImageNet dataset (Le & Yang, 2015).

Our second contribution is an analysis of why implicitly or explicitly regularizing $\mathrm{Tr}(\mathbf{F})$ impacts generalization. We reveal two effects of implicit or explicit regularization of $\mathrm{Tr}(\mathbf{F})$: (1) penalizing $\mathrm{Tr}(\mathbf{F})$ discourages memorizing noisy labels, (2) small $\mathrm{Tr}(\mathbf{F})$ in the early phase of training biases optimization towards a flat minimum, as characterized by the trace of the Hessian.

## 2 IMPLICIT AND EXPLICIT REGULARIZATION OF THE FIM

**Fisher Information Matrix** Consider a probabilistic classification model $p_{\boldsymbol{\theta}}(y|\boldsymbol{x})$, where $\boldsymbol{\theta}$ denotes its parameters. Let $\ell(\boldsymbol{x}, y; \boldsymbol{\theta})$ be the cross-entropy loss function calculated for input $\boldsymbol{x}$ and label $y$. Let $g(\boldsymbol{x}, y; \boldsymbol{\theta}) = \frac{\partial}{\partial \theta} \ell(\boldsymbol{x}, y; \boldsymbol{\theta})$ denote the gradient computed for an example $(\boldsymbol{x}, y)$. The central object that we study is the Fisher Information Matrix $\mathbf{F}$ defined as

$$\mathbf{F}(\boldsymbol{\theta}) = \mathbb{E}_{x \sim \mathcal{X}, \hat{y} \sim p_{\theta}(y|\boldsymbol{x})} [g(\boldsymbol{x}, \hat{y}) g(\boldsymbol{x}, \hat{y})^T], \tag{1}$$

where the expectation is often approximated using the empirical distribution $\hat{\mathcal{X}}$ induced by the training set. We denote its trace by $\mathrm{Tr}(\mathbf{F})$. Later, we also look into the Hessian $\mathbf{H}(\boldsymbol{\theta}) = \frac{\partial^2}{\partial \boldsymbol{\theta}^2} \ell(\boldsymbol{x}, y; \boldsymbol{\theta})$. We denote its trace by $\mathrm{Tr}(\mathbf{H})$.

The FIM can be seen as an approximation to the Hessian (Martens, 2020). In particular, as $p(y|\boldsymbol{x}; \theta) \rightarrow \hat{p}(y|\boldsymbol{x})$, where $\hat{p}(y|\boldsymbol{x})$ is the empirical label distribution, the FIM converges to the

Hessian. Thomas et al. (2020) showed on image classifications tasks that $\text{Tr}(\mathbf{H}) \approx \text{Tr}(\mathbf{F})$ along the optimization trajectory, which we also evidence in Appendix F.

**Fisher Penalty** Several studies have presented evidence that the early phase has a drastic effect on the trajectory in terms of the local curvature of the loss surface (Achille et al., 2019; Jastrzębski et al., 2019; Gur-Ari et al., 2018; Lewkowycz et al., 2020; Leclerc & Madry, 2020). In particular, Lewkowycz et al. (2020); Jastrzębski et al. (2019) show that using a large learning rate in stochastic gradient descent biases training towards low curvature regions of the loss surface very early in training. For example, using a large learning rate in SGD was shown to result in a rapid decay of $\text{Tr}(\mathbf{H})$ along the optimization trajectory Jastrzębski et al. (2019).

Our main contribution is to propose and investigate a specific mechanism by which using a large learning rate or a small batch size implicitly influences final generalization. Our first insight is to shift the focus from studying the Hessian, to studying properties of the FIM. Concretely, we hypothesize that using a large learning rate or a small batch size improves generalization by implicitly penalizing $\text{Tr}(\mathbf{F})$ from the very beginning of training.

The benefit of studying the FIM is that it can be directly and efficiently manipulated during training. In order to study the effect of implicit regularization of $\text{Tr}(\mathbf{F})$, we introduce a regularizer, which we refer to as Fisher penalty, explicitly penalizing $\text{Tr}(\mathbf{F})$. We derive this regularizer in the following way. First, we note that $\text{Tr}(\mathbf{F})$ can be written as $\text{Tr}(\mathbf{F}) = \mathbb{E}_{x \sim \mathcal{X}, \hat{y} \sim p_\theta(y|\boldsymbol{x})} \left[ \| \frac{\partial}{\partial \theta} \ell(\boldsymbol{x}, \hat{y}) \|_2^2 \right]$.

To regularize $\text{Tr}(\mathbf{F})$, we add the following term to the loss function:

$$\ell'(\boldsymbol{x}_{1:B}, y_{1:B}; \boldsymbol{\theta}) = \frac{1}{B} \sum_{i=1}^{B} \ell(\boldsymbol{x}_i, y_i; \boldsymbol{\theta}) + \alpha \left\| \frac{1}{B} \sum_{i=1}^{B} g(\boldsymbol{x}_i, \hat{y}_i) \right\|^2, \qquad (2)$$

where $(\boldsymbol{x}_{1:B}, y_{1:B})$ is a mini-batch, $\hat{y}_i$ is sampled from $p_\theta(y|\boldsymbol{x}_i)$, and $\alpha$ is a hyperparameter. We refer to this regularizer as Fisher penalty. The formulation is based on the empirical observation that $\left\| \frac{1}{B} \sum_{i=1}^{B} g(\boldsymbol{x}_i, \hat{y}_i) \right\|^2$ and $\text{Tr}(\mathbf{F})$ correlate well during training. Crucially, this allows us to reduce the added computational cost of Fisher penalty to that of a single additional backpropagation call (Drucker & Le Cun, 1992). Finally, we compute the gradient of the second term only every 10 optimization steps, and in a given iteration use the most recently computed gradient. We discuss these approximations in detail in Appendix C.

**Catastrophic Fisher Explosion** To illustrate the concepts mentioned in this section, we train a Wide ResNet model (depth 44, width 3) (Zagoruyko & Komodakis, 2016) on the TinyImageNet dataset with SGD and two different learning rates. We illustrate in Figure 1 that the small learning rate leads to dramatic overfitting, which coincides with a sharp increase in $\text{Tr}(\mathbf{F})$ in the early phase of training. We also show in Appendix D that these effects cannot be explained by the difference in learning speed between runs with smaller and learning rates. We call this phenomenon the catastrophic Fisher explosion.

## 3 EARLY-PHASE $\text{Tr}(\mathbf{F})$ CORRELATES WITH FINAL GENERALIZATION

Using a large learning rate ($\eta$) or a small batch size ($S$) in SGD steers optimization to a lower curvature region of the loss surface. However, it remains a hotly debated topic whether this explains their strong regularization effect (Dinh et al., 2017; He et al., 2019; Maddox et al., 2020; Tsuzuku et al., 2019; Yoshida & Miyato, 2017). We begin by studying the connection between $\text{Tr}(\mathbf{F})$ and generalization in experiments across which we vary $\eta$ or $S$ in SGD.

**Experimental setup** We run our experiments in two settings: (1) ResNet-18 with Fixup He et al. (2015); Zhang et al. (2019) trained on the ImageNet dataset (Deng et al., 2009), (2) ResNet-26 initialized with Arpit et al. (2019) trained on the CIFAR-10 and CIFAR-100 datasets (Krizhevsky, 2009). We train each architecture using SGD, with various values of $\eta$, $S$, and random seed.

We define $\text{Tr}(\mathbf{F_i})$ as $\text{Tr}(\mathbf{F})$ during the initial phase of training. The early-phase $\text{Tr}(\mathbf{F})$ is measured when the training loss crosses a task-specific threshold $\epsilon$. For ImageNet, we use learning rates 0.001,

Table 1: Using a 10-30x smaller learning rate (Baseline) results in up to 9% degradation in test accuracy on popular image classification benchmarks (c.f. to optimal $\eta^*$). Adding Fisher penalty (FP) substantially improves generalization and closes the gap to $\eta^*$. We do not use data augmentation with CIFAR-10 and CIFAR-100 to ensure that using a small learning rate does not lead to under-fitting.

| Setting | $\eta^*$ | Baseline | $GP_x$ | GP | FP | $GP_r$ |
|---|---|---|---|---|---|---|
| WResNet/TinyImageNet (aug.) | 54.67% | 52.57% | 52.79% | 56.44% | **56.73%** | 55.41% |
| DenseNet/C100(w/o aug.) | 66.09% | 58.51% | 62.12% | 64.42% | **66.41%** | 66.39% |
| VGG11/C100 (w/o aug.) | 45.86% | 36.86% | 45.26% | 47.35% | **49.87%** | 48.26% |
| WResNet/C100 (w/o aug.) | 53.96% | 46.38% | **58.68%** | 57.68% | 57.05% | 58.15% |
| SimpleCNN/C10(w/o aug.) | 76.94% | 71.32% | 75.68% | 75.73% | 79.66% | **79.76%** |

0.01, 0.1, and $\epsilon = 3.5$. For CIFAR-10, we use learning rates 0.007, 0.01, 0.05, and $\epsilon = 1.2$. For CIFAR-100, we use learning rates 0.001, 0.005, 0.01, and $\epsilon = 3.5$. In all cases, training loss reaches $\epsilon$ between 2 and 7 epochs across different hyper-parameter settings. We repeat similar experiments for different batch sizes in Appendix A.1. The remaining training details can be found in Appendix G.1.

**Results** Figure 2 shows the correlation between $\text{Tr}(\mathbf{F_i})$ and test accuracy across runs with different learning rates. We show results for CIFAR-10 and CIFAR-100 when varying the batch size in Figure 7 in the Appendix. We find that $\text{Tr}(\mathbf{F_i})$ correlates well with the final generalization in our setting, which provides initial evidence for the importance of $\text{Tr}(\mathbf{F})$. It also serves as a stepping stone towards developing a more granular understanding of the role of implicit regularization of $\text{Tr}(\mathbf{F})$ in the following sections.

## 4 FISHER PENALTY

To better understand the significance of the identified correlation between $\text{Tr}(\mathbf{F_i})$ and generalization, we now run experiments in which we directly penalize $\text{Tr}(\mathbf{F})$. We focus our attention on the identified effect of high learning rate on $\text{Tr}(\mathbf{F})$.

**Experimental setting** We use a similar setting as in the previous section, but we include larger models. We run experiments using Wide ResNet (Zagoruyko & Komodakis, 2016) (depth 44 and width 3, with or without BN layers), SimpleCNN (without BN layers), DenseNet (L=40, K=12) (Huang et al., 2017) and VGG-11 (Simonyan & Zisserman, 2015). We train these models on either the CIFAR-10 or the CIFAR-100 datasets. Due to larger computational cost, we replace ImageNet with the TinyImageNet dataset (Le & Yang, 2015) in these experiments.

To investigate if the correlation of $\text{Tr}(\mathbf{F_i})$ and final generalization holds more generally, we apply Fisher penalty in two settings. First, we use a learning rate 10-30x smaller than the optimal one, which both incur up to 9% degradation in test accuracy and results in large value of $\text{Tr}(\mathbf{F_i})$. We also remove data augmentation from the CIFAR-10 and the CIFAR-100 datasets to ensure that training with small learning rate does not result in underfitting. In the second setting, we add Fisher penalty in training with an optimized learning rate using grid search ($\eta^*$) and train with data augmentation.

Fisher penalty penalizes the gradient norm computed using labels sampled from $p_\theta(y|\boldsymbol{x})$. We hypothesize that a similar, but weaker, effect can be introduced by other gradient norm regularizers. First, we compare FP to penalizing the input gradient norm $\|\boldsymbol{g}_x\| = \frac{\partial}{\partial \boldsymbol{x}} \ell(\boldsymbol{x}, y)$, which we denote by $GP_x$ (Varga et al., 2018; Rifai et al., 2011; Drucker & Le Cun, 1992). We also experiment with penalizing the vanilla mini-batch gradient Gulrajani et al. (2017), which we denote by GP. Finally, we experiment with penalizing the mini-batch gradient computed with random labels $\|\boldsymbol{g}_r\| = \frac{\partial}{\partial \boldsymbol{x}} \ell(\boldsymbol{x}, \hat{y})$ where $\hat{y}$ is sampled from a uniform distribution over the label set ($GP_r$). We are not aware of any prior work using GP or $GP_r$ in supervised training, with the exception of Alizadeh et al. (2020) where the authors penalized $\ell_1$ norm of gradients to compress the network towards the end of training.

We tune the hyperparameters on the validation set. More specifically for $\alpha$, we test 10 different values spaced uniformly between $10^{-1} \times v$ to $10^1 \times v$ on a logarithmic scale with $v \in \mathbb{R}_+$. For TinyImageNet

Table 2: Fisher penalty (FP) improves generalization in 4 out of 5 settings when applied with the optimal learning rate $\eta^*$ and trained using standard data augmentation. In 3 out of 5 settings the difference between FP and $\eta^*$ is small (below 1%), which is expected given that FP is aimed at reproducing the regularization effect of large $\eta$, and we compare to training with the optimal $\eta^*$.

| Setting | $\eta^*$ | FP |
|---|---|---|
| DenseNet/C100 (aug.) | **74.41±0.47%** | 74.19±0.51% |
| VGG11/C100 (aug.) | 59.82±1.23% | **65.08±0.53%** |
| WResNet/C100 (aug.) | 69.48±0.30% | **71.53±1.22%** |
| SimpleCNN/C10 (aug.) | 87.16±0.16% | **87.52±0.50%** |
| WResNet/TinyImageNet (aug.) | 54.70±0.04% | **60.00±0.07%** |

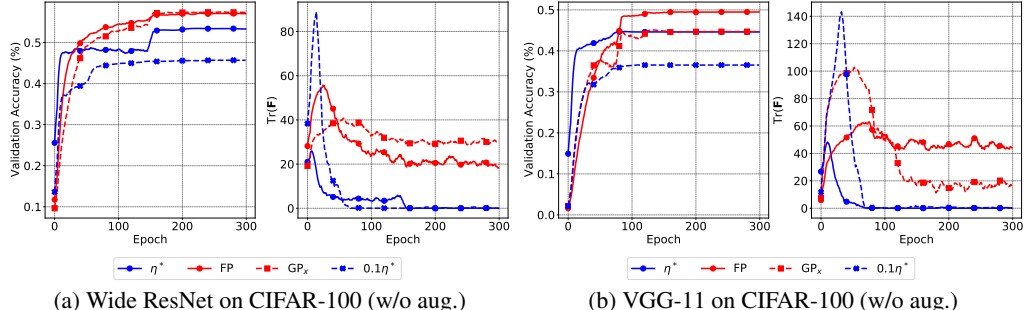

    (a) Wide ResNet on CIFAR-100 (w/o aug.)       (b) VGG-11 on CIFAR-100 (w/o aug.)

Figure 3: Training with FP or GP$_x$ improves generalization and limits early peak of $\mathrm{Tr}(\mathbf{F})$. Each subfigure shows validation accuracy (left) and $\mathrm{Tr}(\mathbf{F})$ (right) for training with $\eta^*$ or a small learning rate (blue) and for training with either GP$_x$ or FP (red). Curves were smoothed for clarity.

we test 5 alternatives instead. To pick the optimal learning rate, we evaluate 5 values spaced equally on a logarithmic scale. We include the remaining experimental details in the Appendix G.2.

**Fisher Penalty improves generalization**    Table 1 summarizes the results of the main experiment. First, we observe that a suboptimal learning rate (10-30x lower than the optimal) leads to dramatic overfitting. We observe a degradation of up to 9% in test accuracy, while achieving perfect training accuracy (see Table 6 in the Appendix).

Fisher penalty closes the gap in test accuracy between the small and optimal learning rate, and even achieves better performance than the optimal learning rate. A similar performance was observed when minimizing $\|g_r\|$. We will come back to this observation in the next section.

GP and GP$_x$ reduce the early value of $\mathrm{Tr}(\mathbf{F})$ (see Table 4 in the Appendix). They, however, generally perform worse than $\mathrm{Tr}(\mathbf{F})$ or GP$_r$ and do not fully close the gap between small and optimal learning rate. We hypothesize they improve generalization by a similar but less direct mechanism than $\mathrm{Tr}(\mathbf{F})$ and GP$_r$.

In the second experimental setting, we apply FP to a network trained with the optimal learning rate $\eta^*$. According to Table 2, Fisher Penalty improves generalization in 4 out of 5 settings. The gap between the baseline and FP is small in 3 out of 5 settings (below 1%), which is natural given that we already regularize training implicitly by using the optimal $\eta$ and data augmentation.

**Geometry and generalization in the early phase of training**    Here, we investigate the temporal aspect of Fisher Penalty on CIFAR-10 and CIFAR-100. In particular, we study whether early penalization of $\mathrm{Tr}(\mathbf{F})$ matters for final generalization.

First, we observe that all gradient-norm regularizers reduce the early value of $\mathrm{Tr}(\mathbf{F})$ closer to $\mathrm{Tr}(\mathbf{F})$ achieved when trained with the optimal learning rate $\eta^*$. We show this effect with Wide ResNet and VGG-11 on CIFAR-100 in Figure 3, and for other experimental settings in the Appendix. We also tabulate the maximum achieved values of $\mathrm{Tr}(\mathbf{F})$ over the optimization trajectory in Appendix A.2.

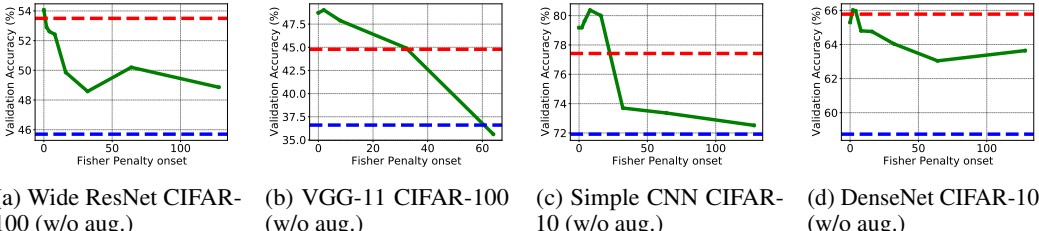

(a) Wide ResNet CIFAR-100 (w/o aug.)  (b) VGG-11 CIFAR-100 (w/o aug.)  (c) Simple CNN CIFAR-10 (w/o aug.)  (d) DenseNet CIFAR-100 (w/o aug.)

Figure 4: Each subplot summarizes an experiment in which we apply Fisher Penalty starting from a certain epoch (x axis) and measure the final test accuracy (y axis). Fisher Penalty has to be applied from the beginning of training to close the generalization gap to the optimal learning rate (c.f. the red horizontal line to the blue horizontal line).

To test the importance of explicitly penalizing $\mathrm{Tr}(\mathbf{F})$ early in training, we start applying it after a certain number of epoch $E \in \{1, 2, 4, 8, 16, 32, 64, 128\}$. We use the best hyperparameter set from the previous experiments. Figure 4 summarizes the results. For both datasets, we observe a consistent pattern. When FP is applied starting from a later epoch, final generalization is significantly worse, and the generalization gap arising from a suboptimal learning rate is not closed.

## 4.1 FISHER PENALTY REDUCES MEMORIZATION

It is not self-evident how regularizing $\mathrm{Tr}(\mathbf{F})$ influences generalization. In this section, we provide evidence that regularizing $\mathrm{Tr}(\mathbf{F})$ slows down learning on data with noisy labels. To study this, we replace labels of the examples in the CIFAR-100 dataset (25% or 50% of the training set) with labels sampled uniformly. While label noise in real datasets is not uniform, methods that perform well with uniform label noise generally are more robust to label noise in real datasets (Jiang et al., 2020a). We also know that datasets such as CIFAR-100 contain many labeling errors (Song et al., 2020). As such, examining how $\mathrm{Tr}(\mathbf{F})$ reduces memorization of synthetic label noise provides an insight into how it improves generalization in our prior experiments.

We expect FP to reduce memorization. When the predictive distribution $p_{\boldsymbol{\theta}}(y|\boldsymbol{x})$ and the true label distribution $p^*(y|\boldsymbol{x})$ are both uniform, $\mathrm{Tr}(\mathbf{F})$ of the specific example $\boldsymbol{x}$ is equivalent to the squared loss gradient norm of the sample example. The proposed Fisher penalty thus minimizes the contribution of the loss gradient from the training examples whose labels were sampled uniformly. In other words, the Fisher penalty implicitly suppresses learning noisy examples, under the assumption that clean examples' label distributions are not uniform.

To study whether the above happens in practice, we compare FP to $\mathrm{GP_x}$, $\mathrm{GP_r}$, and mixup (Zhang et al., 2018). While mixup is not the state-of-the-art approach to learning with noisy labels, it is competitive among approaches that do not require additional data nor multiple stages of training. In particular, it is a component in several state-of-the-art approaches (Li et al., 2020; Song et al., 2020). For gradient norm based regularizers, we evaluate 6 different hyperparameter values spaced uniformly on a logarithmic scale, and for mixup we evaluate $\beta \in \{0.2, 0.4, 0.8, 1.6, 3.2, 6.4\}$. We experiment with the Wide ResNet and VGG-11 models. We describe remaining experimental details in the Appendix G.3.

**Results** We begin by studying the learning dynamics on data with noisy labels through the lens of training accuracy and mini-batch gradient norm. We show the results for VGG-11 and ResNet-50 in Figure 5 and Figure 9 in the Appendix. We observe that FP limits the ability of the model to memorize data more strongly than it limits its ability to learn from clean data. We can further confirm our interpretation of the effect $\mathrm{Tr}(\mathbf{F})$ has on training by studying the gradient norms. As visible in Figure 5, the gradient norm on examples with noisy labels is larger than on clean examples, and the ratio is closer to 1 when large regularization is applied.

We report test accuracy (at the best validation point) in Table 3. We observe that $\mathrm{Tr}(\mathbf{F})$ reduces memorization competitively to mixup. Furthermore, FP performs similarly to $\mathrm{GP_r}$, which agrees with our interpretation of why FP limits learning on examples with noisy labels.

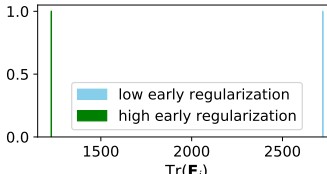 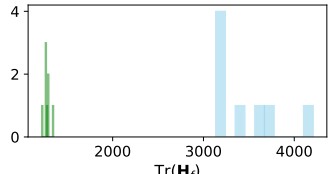 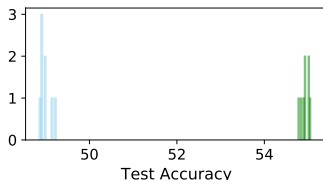

Figure 6: Small $\mathrm{Tr}(\mathbf{F})$ during the early phase of training is more likely to reach wider minima. Left: two ResNet-56 models are trained with two different levels of regularization for 20 epochs on CIFAR-100. $\mathrm{Tr}(\mathbf{F})$ at the end of 20 epochs ($\mathrm{Tr}(\mathbf{F_i})$) is shown. Middle: Each model is then continued trained using the low regularization configuration with different random seeds. A histogram of $\mathrm{Tr}(\mathbf{H})$ at best test accuracy along the trajectory ($\mathrm{Tr}(\mathbf{H_f})$) is shown. Right: a histogram of test accuracy.

Table 3: Fisher Penalty (FP) and $\mathrm{GP_r}$ both reduce memorization competitively to mixup. We measure test accuracy at the best validation point in training with either 25% or 50% examples with noisy labels in the CIFAR-100 dataset.

| Noise | Setting | Baseline | Mixup | $\mathrm{GP_x}$ | FP | $\mathrm{GP_r}$ |
|-------|---------|----------|-------|------|-----|------|
| 25% | VGG-11/C100 | 41.74% | 52.31% | 45.94% | **60.18%** | 58.46% |
|  | ResNet-52/C100 | 53.30% | **61.61%** | 52.70% | 58.31% | 57.60% |
| 50% | VGG-11/C100 | 30.05% | 39.15% | 34.26% | **51.33%** | 50.33% |
|  | ResNet-52/C100 | 43.35% | **51.71%** | 42.99% | 47.99% | 50.08% |

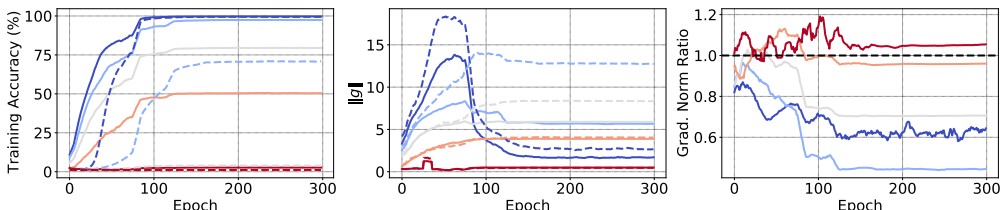

Figure 5: Fisher penalty slows down training on data with noisy labels more strongly than it slows down training on clean data for VGG-11 on CIFAR-100. This likely happens because FP penalizes more strongly gradient norm on data with noisy labels. Left plot shows the training accuracy on examples with clean/noisy labels (solid/dashed line). Middle plot shows the gradient norm evaluated on examples with clean/noisy labels (solid/dashed). Right plot shows the ratio of gradient norm on clean to noisy data. Red to blue color represents the regularization coefficient (from $10^{-2}$ to $10^1$).

## 5 EARLY $\mathrm{Tr}(\mathbf{F})$ INFLUENCES FINAL CURVATURE

To provide further insight why it is important to regularize $\mathrm{Tr}(\mathbf{F})$ during the early phase of training, we establish a connection between the early phase of training and the wide minima hypothesis (Hochreiter & Schmidhuber, 1997; Keskar et al., 2017) which states that *flat* minima *typically* correspond to better generalization. Here, we use $\mathrm{Tr}(\mathbf{H})$ as a measure of flatness.

**Experimental setting** We investigate how likely it is for an optimization trajectory to end up in a wide minimum in two scenarios. 1) When optimization exhibits small $\mathrm{Tr}(\mathbf{F})$ early on. 2) When optimization exhibits large $\mathrm{Tr}(\mathbf{F})$ early on. Specifically, we train two separate ResNet-26 models for 20 epochs using high and low regularization configurations. At epoch 20 we record $\mathrm{Tr}(\mathbf{F})$ for each model. We then use these two models as initialization for 8 separate models each, and continue training using the low regularization configuration with different random seeds. The motivation behind this experiment is to show that the degree of regularization in the early phase biases the model towards minima with certain flatness ($\mathrm{Tr}(\mathbf{H})$) even though no further high regularization

configurations are used during the rest of the training. For all these runs, we record the best test accuracy along the optimization trajectory along with $\text{Tr}(\mathbf{H})$ at the point corresponding to the best test accuracy. We describe the remaining experimental details in Appendix G.4.

**Results** We present the result in Figure 6 for the CIFAR-100 datasets, and for CIFAR-10 in Appendix A.4. A training run that shows a lower $\text{Tr}(\mathbf{F})$ during the early phase is more likely to end up in a wider minimum as opposed to one that reaches large $\text{Tr}(\mathbf{F})$ during the early phase. This happens despite that the late phases of both sets of models use the low regularization configuration. The latter runs have a high variance in the best test accuracy and always end up in sharper minima. In Appendix G.4 we also show evolution of $\text{Tr}(\mathbf{H})$ throughout training, which suggests that this behavior can be attributed to curvature stabilization happening early during training.

## 6 RELATED WORK

SGD's implicit regularization effect has been argued to be a critical component of the empirical success of DNNs (Neyshabur, 2017; Zhang et al., 2016). Much of it is attributed to the choice of hyperparameters (Keskar et al., 2017; Smith & Le, 2018; Jastrzebski et al., 2017), the low complexity bias induced by gradient descent (Xu, 2018; Jacot et al., 2018; Hu et al., 2020) or the cross-entropy loss function (Poggio et al., 2018; Soudry et al., 2018). However, a more mechanistic understanding of how SGD implicitly regularizes DNNs remains a largely unsolved problem.

Prior work on replicating SGD's implicit regularization focused mainly on the loss curvature at the final minimum (Hochreiter & Schmidhuber, 1997). Chaudhari et al. (2019) propose a Langevin dynamics based algorithm for finding update directions that point towards wide minima. Wen et al. (2018) propose to find wide minima by averaging gradients at the neighborhood of the current parameter state. In contrast, we shift the focus to the FIM and the early phase of training. This new perspective allows us to more directly test our theory by explicitly penalizing $\text{Tr}(\mathbf{F})$.

Penalizing $\text{Tr}(\mathbf{F})$ is related to regularizing the input gradient norm, which was shown to be an effective regularizer for deep neural networks (Drucker & Le Cun, 1992; Varga et al., 2018). Chatterjee (2020); Fort et al. (2020) show that SGD avoids memorization by extracting commonalities between examples due to following gradient descent directions shared between examples. Our work is complementary. We argue that SGD implicitly penalizes $\text{Tr}(\mathbf{F})$, which also reduces memorization. Concurrently, Barrett & Dherin (2020) show that SGD implicitly penalizes the gradient norm for large learning rates and propose GP as an explicit regularizer. Similarly, we found that SGD implicitly regularizes $\text{Tr}(\mathbf{F})$, which is the squared gradient norm under labels sampled from $p_{\boldsymbol{\theta}}(y|\boldsymbol{x})$. In contrast to them, we connected the implicit regularization effect of SGD to large curvature in the early phase. We also found GP to be a generally less effective regularizer than FP.

## 7 CONCLUSION

Inspired by recent findings of rapid changes to the local curvature of the loss surface that happen in the early phase of training (Achille et al., 2019; Jastrzębski et al., 2019; Lewkowycz et al., 2020), we investigated more closely the connection between the loss geometry in the early phase of training of neural networks and generalization.

We proposed and investigated a hypothesis that SGD influences generalization by implicitly penalizing the trace of the Fisher Information Matrix ($\text{Tr}(\mathbf{F})$) from the very beginning of training. We show that (1) the value of early $\text{Tr}(\mathbf{F})$ correlates with final generalization, and (2) explicitly regularizing $\text{Tr}(\mathbf{F})$ can substantially improve generalization.

To gain further insight into the mechanism by which penalizing $\text{Tr}(\mathbf{F})$ improves generalization, we investigated training on noisy data. We found that penalizing $\text{Tr}(\mathbf{F})$ reduces memorization by penalizing examples with noisy labels more strongly than clean ones, which seems to happen because it penalizes more strongly their gradient norm. This sheds new light onto implicit regularization effects in SGD, and suggests the utility of penalizing $\text{Tr}(\mathbf{F})$ as an explicit regularizer.

An interesting topic for the future is to put our findings in the context of transfer and continual learning. We hypothesize that catastrophic Fisher explosion (the initial growth of $\text{Tr}(\mathbf{F})$ to a large value) can negatively impact not only generalization, but also transferability of the model.

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

APPPENDIX

## A   ADDITIONAL RESULTS

### A.1   EARLY PHASE $\mathrm{Tr}(\mathbf{F})$ CORRELATES WITH FINAL GENERALIZATION

In this section, we present the additional experimental results for Section 3. The experiments with varying batch size for CIFAR-100 and CIFAR-10 are shown in Figure 7. The conclusions are the same as discussed in the main text in Section 3.

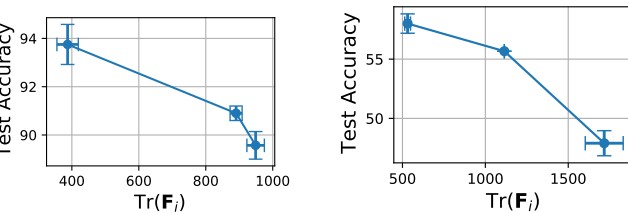

(a) CIFAR-10 (w/ augmentation)   (b) CIFAR-100 (w/o augmentation)

Figure 7: Association between early phase values of $\mathrm{Tr}(\mathbf{F})$ and generalization, holds on the CIFAR-10 and the CIFAR-100 datasets. Each point corresponds to multiple runs with randomly chosen seeds and a specific value of batch size. $\mathrm{Tr}\mathbf{F}_i$ is recorded during early phase (2-7 epochs, see main text for details), while the test accuracy is the maximum value along the entire optimization path (averaged across runs with the same batch size). The horizontal and vertical error bars show the standard deviation of values across runs. The plots show that early phase $\mathrm{Tr}(\mathbf{F})$ is predictive of final generalization.

### A.2   FISHER PENALTY

We first show additional metrics for experiments summarized in Table 1. In Table 6 we show the final training accuracy. Table 4 confirms that generally all gradient norm regularizers reduce the maximum value of $\mathrm{Tr}(\mathbf{F})$ (we measure $\mathrm{Tr}(\mathbf{F})$ starting from after one epoch of training because $\mathrm{Tr}(\mathbf{F})$ explodes in networks with batch normalization layers at initialization). Finally, Table 5 confirms that the regularizers incurred a relatively small additional computational cost.

Figure 8 is a counterpart of Figure 3 for the other two models on the CIFAR-10 and the CIFAR-100 datasets.

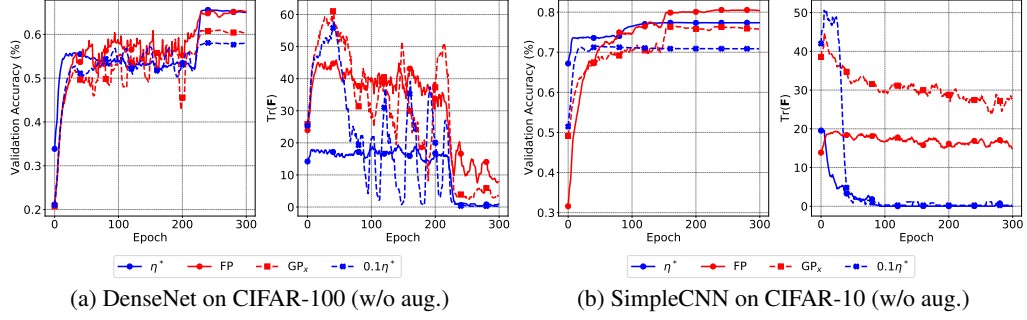

(a) DenseNet on CIFAR-100 (w/o aug.)   (b) SimpleCNN on CIFAR-10 (w/o aug.)

Figure 8: Same as Figure 3, but for DenseNet on CIFAR-100, and SimpleCNN on CIFAR-10. Curves were smoothed for visual clarity.

Table 4: The maximum value of $\mathrm{Tr}(\mathbf{F})$ along the optimization trajectory for experiments on CIFAR-10 or CIFAR-100 included in Table 1.

| Setting | $\eta^*$ | Baseline | GP$_\mathrm{x}$ | GP | FP | GP$_\mathrm{r}$ |
|---|---|---|---|---|---|---|
| DenseNet/C100 (w/o aug.) | 24.68 | 98.17 | 83.64 | 64.33 | 66.24 | 73.66 |
| VGG11/C100 (w/o aug.) | 50.88 | 148.19 | 102.95 | 58.53 | 64.93 | 62.96 |
| WResNet/C100 (w/o aug.) | 26.21 | 91.39 | 41.43 | 40.94 | 56.53 | 39.31 |
| SCNN/C10 (w/o aug.) | 24.21 | 52.05 | 47.96 | 25.03 | 19.63 | 25.35 |

Table 5: Time per epoch (in seconds) for experiments in Table 1.

| Setting | $\eta^*$ | Baseline | GP$_\mathrm{x}$ | GP | FP | GP$_\mathrm{r}$ |
|---|---|---|---|---|---|---|
| WResNet/TinyImageNet (aug.) | 214.45 | 142.69 | 233.14 | 143.78 | 208.62 | 371.74 |
| DenseNet/C100 (w/o aug.) | 78.88 | 57.40 | 77.89 | 78.66 | 97.25 | 75.96 |
| VGG11/C100 (w/o aug.) | 30.50 | 35.27 | 31.54 | 32.52 | 43.41 | 42.40 |
| WResNet/C100 (w/o aug.) | 49.64 | 47.99 | 71.33 | 61.36 | 76.93 | 53.25 |
| SCNN/C10 (w/o aug.) | 18.64 | 19.51 | 26.09 | 19.91 | 21.21 | 20.55 |

Table 6: The final epoch training accuracy for experiments shown in Table 1. Experiments with small learning rate reach no lower accuracy than experiments corresponding to a large learning rate $\eta^*$.

| Setting | $\eta^*$ | Baseline | GP$_\mathrm{x}$ | GP | FP | GP$_\mathrm{r}$ |
|---|---|---|---|---|---|---|
| WResNet/TinyImageNet (aug.) | 99.84% | 99.96% | 99.97% | 93.84% | 81.05% | 86.46% |
| DenseNet/C100 (w/o aug) | 99.98% | 99.97% | 99.96% | 99.91% | 99.91% | 99.39% |
| VGG11/C100 (w/o aug) | 99.98% | 99.98% | 99.85% | 99.62% | 97.73% | 86.32% |
| WResNet/C100 (w/o aug) | 99.98% | 99.98% | 99.97% | 99.96% | 99.99% | 99.94% |
| SCNN/C10 (w/o aug) | 100.00% | 100.00% | 97.79% | 100.00% | 93.80% | 94.64% |

## A.3 FISHER PENALTY REDUCES MEMORIZATION

In this section, we describe additional experimental results for Section 4.1. Figure 9 is the same as Figure 5, but for ResNet-50.

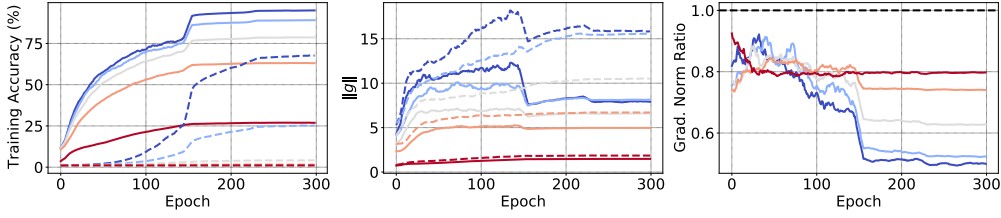

Figure 9: Same as Figure 5, but for ResNet-50.

## A.4 EARLY $\mathrm{Tr}(\mathbf{F})$ INFLUENCES FINAL CURVATURE

In this section, we present additional experimental results for Section 5. The experiment on CIFAR-10 is shown in Figure 10. The conclusions are the same as discussed in the main text in Section 5.

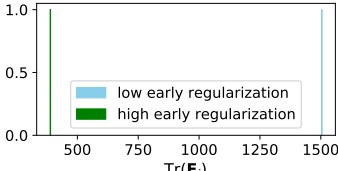 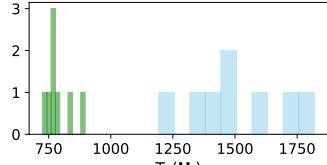 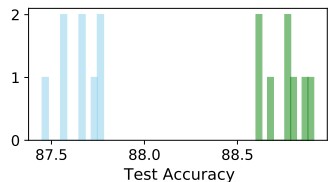

Figure 10: Small $\mathrm{Tr}(\mathbf{F})$ during the early phase of training is more likely to reach wider minima as measured by $\mathrm{Tr}(\mathbf{H})$. Left: 2 models are trained with different levels of regularization for 20 epochs on CIFAR-10. $\mathrm{Tr}(\mathbf{F})$ at the end of 20 epochs (denoted as $\mathrm{Tr}(\mathbf{F_i})$) is shown. Middle: Each model is then used as initialization and trained until convergence using the low regularization configuration with different random seeds. A histogram of $\mathrm{Tr}(\mathbf{H})$ at the point corresponding to the best test accuracy along the trajectory (denoted by $\mathrm{Tr}(\mathbf{H_f})$) is shown. Right: a histogram of the best test accuracy corresponding to middle figure is shown.

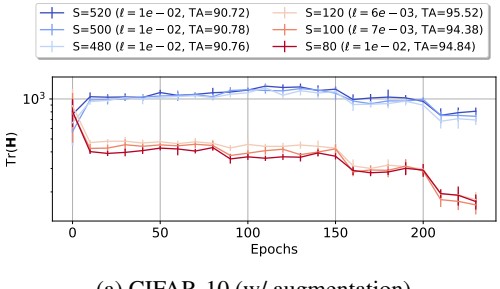

(a) CIFAR-10 (w/ augmentation)

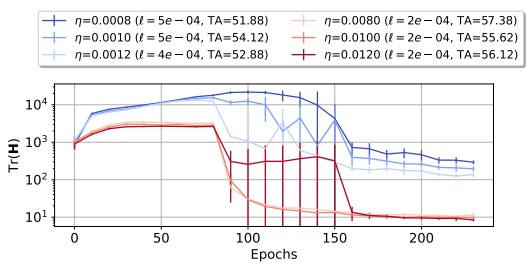

(b) CIFAR-100 (w/o augmentation)

Figure 11: The value of $\mathrm{Tr}(\mathbf{H})$ over the course of training. Each point corresponds to runs with different seeds and a specific value of learning rate $\eta$ and batch size $S$. $\ell$ and TA respectively denote the minimum training loss and the maximum test accuracy along the entire trajectory for the corresponding runs (averaged across seeds). The plots show that flatter optimization trajectories become biased towards flatter minima early during training, at a coarse scale of hyper-parameter values (red vs blue).

Next, to understand why smaller $\mathrm{Tr}(\mathbf{F})$ during early phase is more likely to end up in a wider final minimum, we track $\mathrm{Tr}(\mathbf{H})$ during the entire coarse of training and show that it stabilizes early during training. In this experiment, we create two sets of hyper-parameters: coarse-grained and fine-grained. For CIFAR-10, we use batch size $S \in A \cup B$, where $A = \{480, 500, 520\}$ and $B = \{80, 100, 120\}$. For all batch size configurations, a learning rate of 0.02 is used. Overloading the symbols $A$ and $B$ for CIFAR-100, we use learning rate $\eta \in A \cup B$, where $A = \{0.0008, 0.001, 0.0012\}$ and $B = \{0.008, 0.01, 0.012\}$. For all learning rate configurations, a batch size of 100 is used. In both cases, the elements within each set ($A$ and $B$) vary on a fine-grained scale, while the elements across the two sets vary on a coarse-grained scale. The remaining details and additional experiments can be found in Appendix G.4. The experiments are shown in Figure 11. Notice that after initialization (index 0 on x-axis), the first value is computed at epoch 10 (at which point previous experiments show that entanglement starts to hold with late phase).

We make three observations in this experiment. First, the relative ordering of $\mathrm{Tr}(\mathbf{H})$ values for runs between sets $A$ vs $B$ stay the same after the first 10 epochs. Second, the degree of entanglement is higher between any two epochs when looking at runs across sets $A$ and $B$, while it is weaker when looking at runs within any one the sets. Finally, test accuracies for set $B$ runs are always higher than those of set $A$ runs, but this trend is not strong for runs within any one set. Note that the minimum loss values are roughly at a similar scale for each dataset and they are all at or below $10^{-2}$.

## B    COMPUTATION OF $\mathrm{Tr}(\mathbf{H})$

We computed $\mathrm{Tr}(\mathbf{H})$ in our experiments using the Hutchinson's estimator Hutchinson (1990),

$$
\begin{aligned}
Tr(\mathbf{H}) &= Tr(\mathbf{H} \cdot \mathbf{I}) \\
&= Tr(\mathbf{H} \cdot \mathbb{E}[\mathbf{z}\mathbf{z}^T]) \\
&= \mathbb{E}[Tr(\mathbf{H} \cdot \mathbf{z}\mathbf{z}^T)] \\
&= \mathbb{E}[\mathbf{z}^T \mathbf{H} \cdot \mathbf{z}] \\
&\approx \frac{1}{M} \sum_{i=1}^{M} \mathbf{z}_i^T \mathbf{H} \cdot \mathbf{z}_i \\
&= \frac{1}{M} \sum_{i=1}^{M} \mathbf{z}_i^T \frac{\partial}{\partial \theta} \left( \frac{\partial \ell}{\partial \theta^T} \right) \cdot \mathbf{z}_i \\
&= \frac{1}{M} \sum_{i=1}^{M} \mathbf{z}_i^T \frac{\partial}{\partial \theta} \left( \frac{\partial \ell}{\partial \theta}^T \mathbf{z}_i \right),
\end{aligned}
$$

where $\mathbf{I}$ is the identity matrix, $\mathbf{z}$ is a multi-variate standard Gaussian random variable, and $\mathbf{z}_i$'s are i.i.d. instances of $\mathbf{z}$. The larger the value of $M$, the more accurate the approximation is. We used $M = 30$. To make the above computation efficient, note that the gradient $\frac{\partial \ell}{\partial \theta}$ only needs to be computed once and it can be re-used in the summation over the $M$ samples.

## C    APPROXIMATIONS IN FISHER PENALTY

In this section, we describe the approximations made in Fisher Penalty in detail. Recall, that $\mathrm{Tr}(\mathbf{F})$ can be expressed as

$$
\mathrm{Tr}(\mathbf{F}) = \mathbb{E}_{x \sim \mathcal{X}, \hat{y} \sim p_\theta(y|\boldsymbol{x})} \left[ \| \frac{\partial}{\partial \theta} \ell(\boldsymbol{x}, \hat{y}) \|_2^2 \right]. \tag{3}
$$

In the preliminary experiments, we found empirically that we can use the norm of the expected gradient rather than the expected norm of the gradient, which is a more direct expression of $\mathrm{Tr}(\mathbf{F})$:

$$
\begin{aligned}
\nabla \mathbb{E}_{x \sim \mathcal{X}, \hat{y} \sim p_\theta(y|\boldsymbol{x})} \left[ \left\| \frac{\partial}{\partial \theta} \ell(\boldsymbol{x}, \hat{y}) \right\|_2^2 \right] &\approx \frac{1}{N} \sum_{n=1}^{N} \frac{1}{M} \sum_{m=1}^{M} \nabla \left\| \frac{\partial}{\partial \theta} \ell(\boldsymbol{x}_n, \hat{y}_{nm}) \right\|_2^2 \\
&\geq \nabla \left\| \frac{1}{NM} \sum_{n=1}^{N} \sum_{m=1}^{M} \frac{\partial}{\partial \theta} \ell(\boldsymbol{x}_n, \hat{y}_{nm}) \right\|_2^2,
\end{aligned}
$$

where $N$ and $M$ are the minibatch size and the number of samples from $p_\theta(y|\boldsymbol{x}_n)$, respectively. This greatly improves the computational efficiency. With $N = B$ and $M = 1$, we end up with the following learning objective function:

$$
\ell'(\boldsymbol{x}_{1:B}, y_{1:B}; \boldsymbol{\theta}) = \frac{1}{B} \sum_{i=1}^{B} \ell(\boldsymbol{x}_i, y_i; \boldsymbol{\theta}) + \alpha \left\| \frac{1}{B} \sum_{i=1}^{B} g(\boldsymbol{x}_i, \hat{y}_i) \right\|^2. \tag{4}
$$

We found empirically that $\left\| \frac{1}{B} \sum_{i=1}^{B} g(\boldsymbol{x}_i, \hat{y}_i) \right\|^2$, which we denote by $\mathrm{Tr}(\mathbf{F}^B)$, and $\mathrm{Tr}(\mathbf{F})$ correlate well during training. To demonstrate this, we train SimpleCNN on the CIFAR-10 dataset with 5 different learning rates (from $10^{-3}$ to $10^{-1}$). The outcome is shown in Figure 12. We see that for most of the training, with the exception of the final phase, $\mathrm{Tr}(\mathbf{F}^B)$ and $\mathrm{Tr}(\mathbf{F})$ correlate extremely well. Equally importantly, we find that using a large learning affects both $\mathrm{Tr}(\mathbf{F}^B)$ and $\mathrm{Tr}(\mathbf{F})$, which further suggests the two are closely connected.

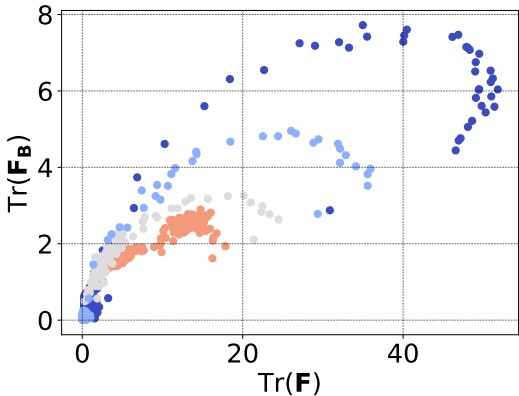

Figure 12: Correlation between $\mathrm{Tr}(\mathbf{F})$ and $\mathrm{Tr}(\mathbf{F}^B)$ for SimpleCNN trained on the CIFAR-10 dataset. Blue to red color denotes learning rates from $10^{-3}$ to $10^{-1}$. The value of $\mathrm{Tr}(\mathbf{F})$ and $\mathrm{Tr}(\mathbf{F}^B)$ correlate strongly for the most of the training trajectory. Using large learning rate reduces both $\mathrm{Tr}(\mathbf{F})$ and $\mathrm{Tr}(\mathbf{F}^B)$.

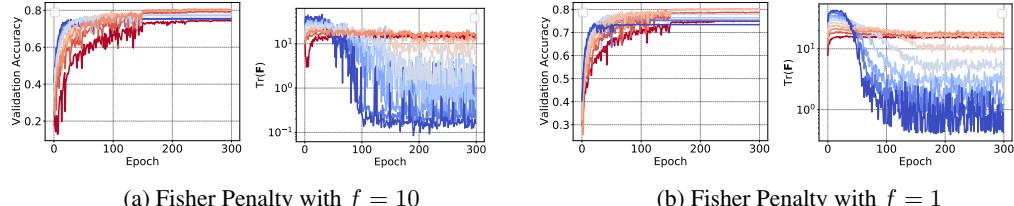

(a) Fisher Penalty with $f = 10$          (b) Fisher Penalty with $f = 1$

Figure 13: A comparison between the effect of recomputing Fisher Penalty gradient every 10 iterations (left) or every iteration (right), with respect to validation accuracy and $\mathrm{Tr}(\mathbf{F})$. We denote by $f$ the frequency with which we update the gradient. Both experiments result in approximately 80% test accuracy with the best configuration.

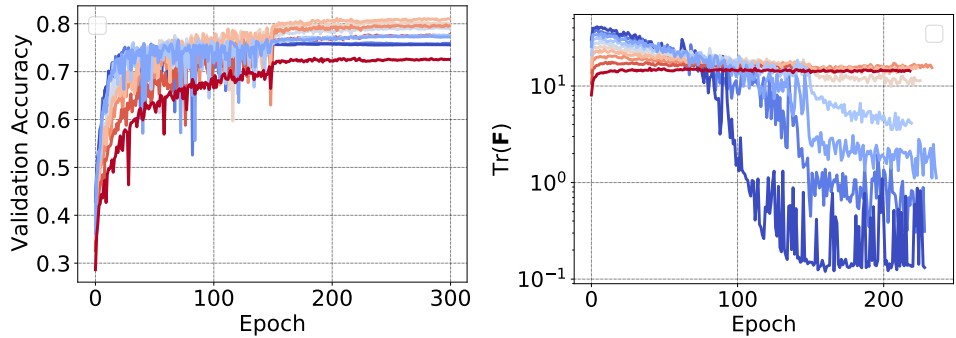

Figure 14: Using Fisher Penalty without the approximation results in a similar generalization performance. We penalize the norm of the gradient rather than norm of the mini-batch gradient (as in Equation 2). We observe that this variant of Fisher Penalty improves generalization to a similar degree as the version of Fisher Penalty used in the paper (c.f. Figure 13.), achieving 79.7% test accuracy.

We also update the gradient of $\mathrm{Tr}(\mathbf{F}^B)$ only every 10 optimization steps. We found empirically it does not affect generalization performance nor the ability to regularize $\mathrm{Tr}(\mathbf{F})$ in our setting. However, we acknowledge that it is plausible that this choice would have to be reconsidered in training with very large learning rates or with larger models.

Figure 13 compares learning curves of training with FP recomputed every optimization step, or every 10 optimization steps. For each, we tune the hyperparameter $\alpha$, checking 10 values equally spaced between $10^{-2}$ and $10^0$ on a logarithmic scale. We observe that for the optimal value of $\alpha$, both validation accuracy and $\text{Tr}(\mathbf{F})$ are similar between the two runs. Both experiments achieve approximately 80% test accuracy.

Finally, to ensure that using the approximation in Equation 2 does not negatively affect how Fisher Penalty improves generalization or reduces the value of $\text{Tr}(\mathbf{F})$, we experiment with a variant of Fisher Penalty without the approximation. Please recall that we always measure $\text{Tr}(\mathbf{F})$ (i.e. we do not use approximations in computing $\text{Tr}(\mathbf{F})$ that is reported in the plots), regardless of what variant of penalty is used in regularizing the training.

Specifically, we augment the loss function with the norm of the gradient computed on the first example in the mini-batch as follows

$$\ell'(\boldsymbol{x}_{1:B}, y_{1:B}; \boldsymbol{\theta}) = \frac{1}{B} \sum_{i=1}^{B} \ell(\boldsymbol{x}_i, y_i; \boldsymbol{\theta}) + \alpha \left\| g(\boldsymbol{x}_1, \hat{y}_1) \right\|^2. \tag{5}$$

We apply this penalty in each optimization step. We tune the hyperparameter $\alpha$, checking 10 values equally spaced between $10^{-4}$ and $10^{-2}$ on a logarithmic scale.

Figure 14 summarizes the results. We observe that the best value of $\alpha$ yields 79.7% test accuracy, compared to 80.02% test accuracy yielded by the Fisher Penalty. The effect on $\text{Tr}(\mathbf{F})$ is also very similar. We observe that the best run corresponds to maximum value of $\text{Tr}(\mathbf{F})$ of 24.16, compared to that of 21.38 achieved by Fisher Penalty. These results suggest that the approximation used in Fisher Penalty only improves the generalization and flattening effects of Fisher Penalty.

## D    A CLOSER LOOK AT THE SURPRISING EFFECT OF LEARNING RATE ON THE LOSS GEOMETRY IN THE EARLY PHASE OF TRAINING

It is intuitive to hypothesize that the catastrophic Fisher explosion (the initial growth of the value of $\text{Tr}(\mathbf{F})$) occurs during training with a large learning rate, but is overlooked due to not sufficiently fine-grained computation of $\text{Tr}(\mathbf{F})$. In this section we show evidence against this hypothesis based on the literature mentioned in the main text. We also run additional experiments in which we compute the value of $\text{Tr}(\mathbf{F})$ at each iteration.

The surprising effect of the learning rate on the geometry of the loss surface (e.g. the value of $\text{Tr}(\mathbf{F})$) was demonstrated in prior works (Jastrzębski et al., 2019; Golatkar et al., 2019; Lewkowycz et al., 2020; Leclerc & Madry, 2020). In particular, Jastrzębski et al. (2020); Lewkowycz et al. (2020) show that training with large learning rate rapidly escapes regions of high curvature, where curvature is understood as the spectral norm of the Hessian evaluated at the current point of the loss surface. Perhaps the most direct experimental data against this hypothesis can be found in Anonymous (2021) in Figure 1, where training with Gradient Descent finds regions of the loss surface with large curvature for small learning rate rapidly in the early phase of training.

We also run the following experiment to provide further evidence against the hypothesis. We train SimpleCNN on the CIFAR-10 dataset using two different learning rates, while computing the value of $\text{Tr}(\mathbf{F})$ for every mini-batch. We use 128 random samples in each iteration to estimate $\text{Tr}(\mathbf{F})$.

We find that training with a large learning rate never (even for a single optimization step) enters a region with the value of $\text{Tr}(\mathbf{F})$ as large as reached during training with a small learning rate. Figure 15 shows the experimental data.

We also found similar to hold when varying the batch size, see Section E, which further shows that the observed effects cannot be explained by the difference in learning speed incurred by using a small learning rate.

To summarize, both the published evidence of Jastrzebski et al. (2020); Lewkowycz et al. (2020); Anonymous (2021), as well as our additional experiments are inconsistent with the hypothesis that the results in this paper can be explained by differences in training speed between experiments using large and small learning rates.

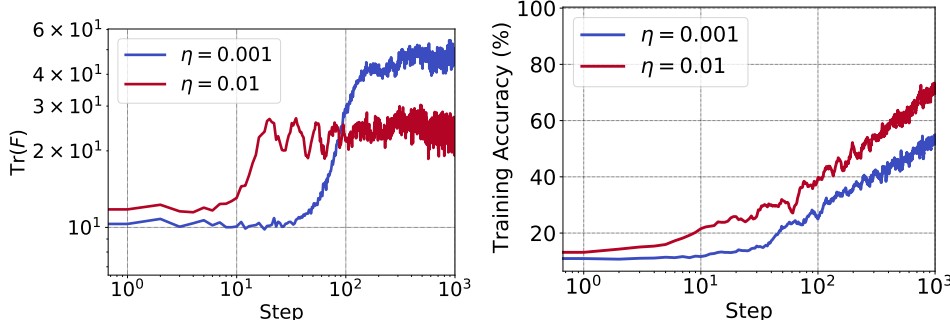

Figure 15: Training with a large learning rate never (even for a single optimization step) enters a region with as large value of $\mathrm{Tr}(\mathbf{F})$ as the maximum value of $\mathrm{Tr}(\mathbf{F})$ reached during training with a small learning rate. We run the experiment using SimpleCNN on the CIFAR-10 dataset with two different learning rates. The left plot shows the value of $\mathrm{Tr}(\mathbf{F})$ computed at each iteration, and the right plot shows training accuracy computed on the current mini-batch (curve has been smoothed for clarity).

# E   CATASTROPHIC FISHER EXPLOSION HOLDS IN TRAINING WITH LARGE BATCH-SIZE

In this section, we show evidence that the conclusions transfer to large batch size training. Namely, we show that (1) catastrophic Fisher explosion also occurs in large batch size training, and (2) Fisher Penalty can improve generalization and close the generalization gap due to using a large batch size (Keskar et al., 2017).

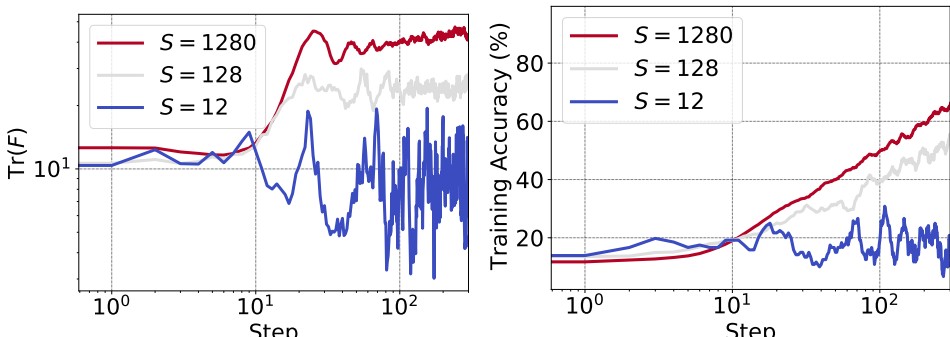

Figure 16: Catastrophic Fisher explosion in large batch size training. Experiment run on the CIFAR-10 and dataset the SimpleCNN model. The left plot shows the value of $\mathrm{Tr}(\mathbf{F})$ computed at each iteration, and the right plot shows training accuracy computed on the current mini-batch (curve has been smoothed for clarity).

We first train SimpleCNN on the CIFAR-10 dataset using three different batch sizes, while computing the value of $\mathrm{Tr}(\mathbf{F})$ for every mini-batch. We use 128 random samples in each iteration to estimate $\mathrm{Tr}(\mathbf{F})$. Figure 16 summarizes the experiment. We observe that training with a large batch size enters a region of the loss surface with a substantially larger value of $\mathrm{Tr}(\mathbf{F})$ than the small batch size.

Next, we run a variant of one of the experiments in Table 1. Instead of using a suboptimal (smaller) learning rate, we use a suboptimal (larger) batch size. Specifically, we train SimpleCNN on the CIFAR-10 dataset (without augmentation) with a 10x larger batch size while keeping learning rate the same. Using a larger batch size results in 3.24% lower test accuracy (76.94% compared to 73.7% test accuracy, c.f. with Table 1).

We next experiment with Fisher Penalty. We apply the penalty in each optimization step and use the first 128 examples when computing the penalty. We also use a 2x lower learning rate, which stabilizes training but does not improve generalization on its own (training with this learning rate reaches 73.59% test accuracy). Figure 17 shows $\mathrm{Tr}(\mathbf{F})$ and validation accuracy during training for different values of the penalty. We observe that Fisher Penalty improves test accuracy from 73.59% to 78.7%. Applying Fisher Penalty also effectively reduces the peak value of $\mathrm{Tr}(\mathbf{F})$/

Taken together, the results suggest that Catastrophic Fisher explosion holds in large batch size training; using a small batch size improves generalization by a similar mechanism as using a large batch size, which can be introduced explicitly in the form of Fisher Penalty.

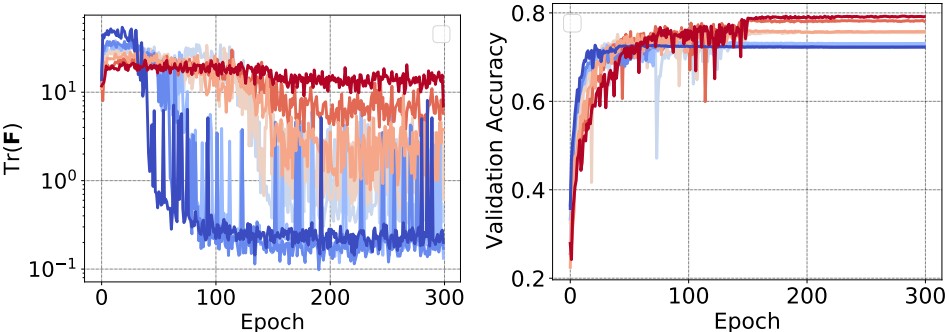

Figure 17: Fisher Penalty improves in large batch size training. Experiment run on the CIFAR-10 dataset (without augmentation) and the SimpleCNN model. Warmer color corresponds to larger coefficient used in Fisher Penalty.

## F $\mathrm{Tr}(\mathbf{H})$ AND $\mathrm{Tr}(\mathbf{F})$ CORRELATE STRONGLY

We demonstrate a strong correlation between $\mathrm{Tr}(\mathbf{H})$ and $\mathrm{Tr}(\mathbf{F})$ for DenseNet, ResNet-56 and SimpleCNN in Figure 18. We calculate $\mathrm{Tr}(\mathbf{F})$ using a mini-batch. We see that $\mathrm{Tr}(\mathbf{F})$ has a smaller magnitude (because we use the mini-batch gradient which has lower variance), but correlates extremely well with $\mathrm{Tr}(\mathbf{H})$.

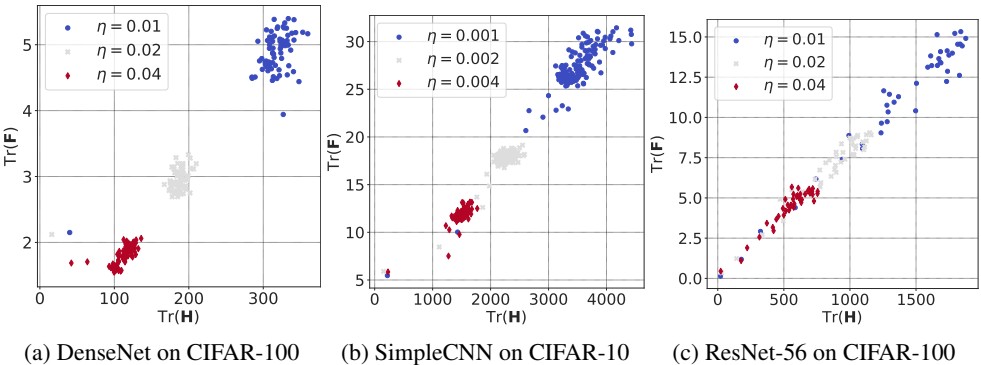

(a) DenseNet on CIFAR-100    (b) SimpleCNN on CIFAR-10    (c) ResNet-56 on CIFAR-100

Figure 18: Correlation between $\mathrm{Tr}(\mathbf{F})$ and $\mathrm{Tr}(\mathbf{H})$.

## G ADDITIONAL EXPERIMENTAL DETAILS

### G.1 EARLY PHASE $\mathrm{Tr}(\mathbf{F})$ CORRELATES WITH FINAL GENERALIZATION

Here, we describe additional details for experiments in Section 3.

In the experiments with batch size, for CIFAR-10, we use batch sizes 100, 500 and 700, and $\epsilon = 1.2$. For CIFAR-100, we use batch sizes 100, 300 and 700, and $\epsilon = 3.5$. These thresholds are crossed between 2 and 7 epochs across different hyperparameter settings. The remaining details for CIFAR-100 and CIFAR-10 are the same as described in main text. The optimization details for these datasets are as follows.

**ImageNet**: No data augmentation was used in order to allow training loss to converge to small values. We use a batch size of 256. Training is done using SGD with momentum set to 0.9, weight decay set to $1e - 4$, and with base learning rate as per the aforementioned details. Learning rate is dropped by a factor of 0.1 after 29 epochs and training is ended at around 50 epochs at which most runs converge to small loss values. No batch normalization is used and weight are initialized using Fixup Zhang et al. (2019). For each hyperparameter setting, we run two experiments with different random seeds due to the computational overhead. We compute $\mathrm{Tr}(\mathbf{F})$ using 2500 samples (similarly to **?**).

**CIFAR-10**: We used random flipping as data augmentation. In the experiments with variation in learning rates, we use a batch size of 256. In the experiments with variation in batch size, we use a learning rate of 0.02. Training is done using SGD with momentum set to 0.9, weight decay set to $1e - 5$, and with base learning rate as per the aforementioned details. Learning rate is dropped by a factor of 0.5 at epochs 60, 120, and 170, and training is ended at 200 epochs at which most runs converge to small loss values. No batch normalization is used and weight are initialized using Arpit et al. (2019). For each hyperparameter setting, we run 32 experiments with different random seeds. We compute $\mathrm{Tr}(\mathbf{F})$ using 5000 samples.

**CIFAR-100**: No data augmentation was used for CIFAR-100 to allow training loss to converge to small values. We used random flipping as data augmentation for CIFAR-10. In the experiments with variation in learning rates, we use a batch size of 100. In the experiments with variation in batch size, we use a learning rates of 0.02. Training is done using SGD with momentum set to 0.9, weight decay set to $1e - 5$, and with base learning rate as per the aforementioned details. Learning rate is dropped by a factor of 0.5 at epochs 60, 120, and 170, and training is ended at 200 epochs at which most runs converge to small loss values. No batch normalization is used and weight are initialized using Arpit et al. (2019). For each hyperparameter setting, we run 32 experiments with different random seeds. We compute $\mathrm{Tr}(\mathbf{F})$ using 5000 samples.

## G.2 FISHER PENALTY

Here, we describe the remaining details for the experiments in Section 4. We first describe how we tune hyperparameters in these experiments. In the remainder of this section, we describe each setting used in detail .

**Tuning hyperparameters**  In all experiments, we refer to the optimal learning rate $\eta^*$ as the learning rate optimized using grid search. In most experiments we check 5 different learning rate values uniformly spaced on a logarithmic scale, usually between $10^{-2}$ and $10^0$. In some experiments we adapt the range to ensure that the range includes the optimal learning rate. We tune the learning rate only once for each configuration (i.e. we do not repeat it for different random seeds).

In the first setting, for most experiments involving gradient norm regularizers, we use $10\times$ smaller learning rate than $\eta^*$. For TinyImageNet, we use $30\times$ smaller learning rate than $\eta^*$. To pick the regularization coefficient $\alpha$, we evaluate 10 different values uniformly spaced on a logarithmic scale between $10^{-1} \times v$ to $10^1 \times v$ with $v \in \mathbb{R}_+$. We choose the best performing $\alpha$ according to best validation accuracy. We pick the value of $v$ manually with the aim that the optimal $\alpha$ is included in this range. We generally found that $v = 0.01$ works well for GP, GP$_\mathrm{r}$, and FP. For GP$_\mathrm{x}$ we found in some experiments that it is necessary to pick larger values of $v$.

**Measuring $\mathrm{Tr}(\mathbf{F})$**  We measure $\mathrm{Tr}(\mathbf{F})$ using the number of examples equal to the batch size used in training. For experiments with Batch Normalization layers, we use Batch Normalization in evaluation mode due to the practical reason that computing $\mathrm{Tr}(\mathbf{F})$ uses batch size of 1, and hence $\mathrm{Tr}(\mathbf{F})$ is not defined for a network with Batch Normalization layers in training mode.

**DenseNet on the CIFAR-100 dataset**  We use the DenseNet (L=40, k=12) configuration from Huang et al. (2017). We largely follow the experimental setting in Huang et al. (2017). We use

the standard data augmentation (where noted) and data normalization for CIFAR-100. We hold out random 5000 examples as the validation set. We train the model using SGD with momentum of 0.9, a batch size of 128, and weight decay of 0.0001. Following Huang et al. (2017), we train for 300 epochs and decay the learning rate by a factor of 0.1 after epochs 150 and 225. To reduce variance, in testing we update Batch Normalization statistics using 100 batches from the training set.

**Wide ResNet on the CIFAR-100 dataset**    We train Wide ResNet (depth 44 and width 3, without Batch Normalization layers). We largely follow experimental setting in He et al. (2015).We use the standard data augmentation and data normalization for CIFAR-100. We hold out random 5000 examples as the validation set. We train the model using SGD with momentum of 0.9, a batch size of 128, weight decay of 0.0010. Following He et al. (2015), we train for 300 epochs and decay the learning rate by a factor of 0.1 after epochs 150 and 225. We remove Batch Normalization layers. To ensure stable training we use the SkipInit initialization (De & Smith, 2020).

**VGG-11 on the CIFAR-100 dataset**    We adapt the VGG-11 model (Simonyan & Zisserman, 2015) to CIFAR-100. We do not use dropout nor Batch Normalization layers. We hold out random 5000 examples as the validation set. We use the standard data augmentation (where noted) and data normalization for CIFAR-100. We train the model using SGD with momentum of 0.9, a batch size of 128, and weight decay of 0.0001. We train the model for 300 epochs, and decay the learning rate by a factor of 0.1 after every 40 epochs starting from epoch 80.

**SimpleCNN on the CIFAR-10 dataset**    We also run experiments on the CNN example architecture from the Keras example repository (Chollet & others, 2015)[1], which we change slightly. Specifically, we remove dropout and reduce the size of the final fully-connected layer to 128. We train it for 300 epochs and decay the learning rate by a factor of 0.1 after the epochs 150 and 225. We train the model using SGD with momentum of 0.9, a batch size of 128.

**Wide ResNet on the TinyImageNet dataset**    We train Wide ResNet (depth 44 and width 3, with Batch Normalization layers) on TinyImageNet Le & Yang (2015). TinyImageNet consists of subset of 100,000 examples from ImageNet that we downsized to $32\times32$ pixels. We train the model using SGD with momentum of 0.9, a batch size of 128, and weight decay of 0.0001. We train for 300 epochs and decay the learning rate by a factor of 0.1 after epochs 150 and 225. We train the model using SGD with momentum of 0.9, a batch size of 128. We do not use validation in TinyImageNet due to its larger size. To reduce variance, in testing we update Batch Normalization statistics using 100 batches from the training set.

### G.3    FISHER PENALTY REDUCES MEMORIZATION

Here, we describe additional experimental details for Section 4.1. We use two configurations described in Section G.2: VGG-11 trained on CIFAR-100 dataset, and Wide ResNe trained on the CIFAR-100 dataset. We tune the regularization coefficient $\alpha$ in the range $\{0.01, 0.1, 0.31, 10\}$, with the exception of $GP_x$ for which we use the range $\{10, 30, 100, 300, 1000\}$. We tuned mixup coefficient in the range $\{0.4, 0.8, 1.6, 3.2, 6.4\}$. We removed weight decay in these experiments.

### G.4    EARLY $\mathrm{Tr}(\mathbf{F})$ INFLUENCES FINAL CURVATURE

**CIFAR-10**: We used random flipping as data augmentation for CIFAR-10. We use a learning rate of 0.02 for all experiments. Training is done using SGD with momentum 0.9, weight decay $1e-5$, and with batch size as shown in figures. Learning rate is drop by a factor of 0.5 at 80, 150, and 200 epochs, and training is ended at 250 epochs. No batch normalization is used and weight are initialized using Arpit et al. (2019). For each batch size, we run 32 experiments with different random seeds. We compute $\mathrm{Tr}(\mathbf{F})$ using 5000 samples.

**CIFAR-100**: No data augmentation is used. We use a batch size of 100 for all experiments. Training is done using SGD with momentum 0.9, weight decay $1e-5$, and with base learning rate as shown in figures. Learning rate is drop by a factor of 0.5 at 80, 150, and 200 epochs, and training is ended at 250 epochs. No batch normalization is used and weight are initialized using Arpit et al. (2019). For

---

[1]Accessible at https://github.com/keras-team/keras/blob/master/examples/cifar10_cnn.py.

each learning rate, we run 32 experiments with different random seeds. We compute $\mathrm{Tr}(\mathbf{F})$ using 5000 samples.

