# OpenReview forum: "Catastrophic Fisher Explosion: Early Phase Fisher Matrix Impacts Generalization"
_ICLR.cc/2021/Conference — Reject_

### Official Review · AnonReviewer2 · 2020-10-25
**Interesting contribution, extensive experiments, some questions**

**Rating:** 6
**Confidence:** 3

**Review:**

This paper empirically investigates the effect of the trace of the Fisher Information Matrix (FIM) early in training has on the generalization of SGD. Authors demonstrate that the effect of optimally chosen learning rate and batch size for SGD can be modeled as an implicit penalty on the trace of FIM. They argue that explicitly penalizing the trace of FIM discourages memorizing noisy labels, thus leading to better generalization. Furthermore, they experimentally show that the early low value of the trace of FIM may bias the optimization towards a flat optimum which has been observed to correlate well with good generalization.

*Positives:*
+ The paper is well-written and easy to follow. Authors convey their message clearly.
+ The paper's motivation is definitely relevant. Understanding the connection between the early stage loss landscape of neural networks and final generalization is crucial and has drawn significant attention recently.
+ The introduced Fisher-penalty is interesting and the authors provide insights into the mechanism how regularizing the trace of FIM might improve generalization.

*Negatives:*
- The exact mechanism how FP influences learning with noisy labels could be investigated a bit more rigorously. The discussion provided in the paper is not completely clear.
- The experiments are somewhat inconclusive in some cases (see below) and would require some extra comments/explanation.

Overall, the paper has some interesting contribution by demonstrating the positive effect of various gradient norm penalties on the generalization of SGD and it is tied well into existing literature. Furthermore, the experiments are comprehensive and in-depth. Therefore I would recommend acceptance if a couple of issues are addressed.

First, as seen in Fig. 4 generalization peaks if FP is applied starting after some positive number of epochs. The difference in validation accuracy between turning on FP from the very beginning and only turning it on somewhat later is not always negligible. This observation somewhat violates the 'earlier we regularize the better' idea. Can the authors comment on this discrepancy?

Second, the benefit of high early regularization is somewhat inconclusive based on Fig. 10. It appears that the best generalization belongs to low early regularization, whereas high early regularization leads to better average test accuracy. I would be interested to know how much FP might hinder optimization by discouraging the exploration of certain directions in the loss landscape. Can it even contribute to trapping the network in local optima by penalizing large gradients needed to escape such 'bad' optima?

Minor comment: there is a typo in the caption of Fig. 6.

---

> ### Author Response · Authors · 2020-11-20
> **Answers to remaining questions**
>
> **” I would be interested to know how much FP might hinder optimization by discouraging the exploration of certain directions in the loss landscape. Can it even contribute to trapping the network in local optima by penalizing large gradients needed to escape such 'bad' optima?”**
>
> &nbsp;&nbsp;&nbsp; In short, it depends on which examples have large gradient norms if one would resample their labels. These examples will learn slower when Fisher Penalty is applied.
>
> There is one specific case in which we could imagine Fisher Penalty (as well as using a large learning rate) to trap the model in a bad solution. If the training examples with noisy (incorrect) labels are easier to learn than the correctly labeled samples. This can happen for instance when (say) all the corrupted samples have the same label.
>
> However, the above scenario generally should not happen in practice, at least not in the synthetic noise scenario. In particular, the authors of [1] show explicitly that examples with incorrect labels on CIFAR-10 tend to have lower gradient norms. As the authors discuss further, it is also supported by the fact that deep neural networks first learn ‘clean’ examples, which makes sense if they have larger gradient norms.
>
> We will include the discussion above in the paper. Please let us know if you have any further questions about this point.
>
> [1] https://openreview.net/forum?id=ryeFY0EFwS
>
> **” The exact mechanism of how FP influences learning with noisy labels could be investigated a bit more rigorously. The discussion provided in the paper is not completely clear.”**
>
> Sorry, it was a bit shortened due to space requirements. We will include a similar discussion as in the answer to your previous question. Please let us know if you have any specific questions.
>
> **“Minor comment: there is a typo in the caption of Fig. 6”**
>
> &nbsp;&nbsp;&nbsp; Thank you.

---

> ### Author Response · Authors · 2020-11-20
> **Thank you for the review**
>
> Thank you for your time and the review! Please find our answers below and let us know should any additional experiments or answers be helpful.
>
> **“Validation accuracy between turning on FP from the very beginning and only turning it on somewhat later is not always negligible. This observation somewhat violates the 'earlier we regularize the better' idea. Can the authors comment on this discrepancy?”**
>
> &nbsp;&nbsp;&nbsp; That’s a very good point. Interestingly, a standard trick of the trade is to gradually increase the learning rate in the early phase of training, which is usually referred to as “warmup”.It is default for training Transformers in NLP, for example.
>
> Hence, applying a large learning rate from the first iteration of training is often suboptimal. We believe the experiments in Figure 4 suggest the same is true for Fisher Penalty, which would be consistent with the core claim of the paper that Fisher Penalty mimics the way large learning rate improves generalization.
>
> More specifically, Figure 4 shows that Fisher Penalty has to be applied within a relatively short window in the beginning of training (~1-3% training time, full training time is between 200 to 300 epochs depending on the setting). If Fisher Penalty is applied later than 1-3% training time, we cannot recover from bad generalization due to using a too small learning rate (which can be read from the graph as the time the curve intersects the horizontal line). It is also worth noting that, as visible in Figure 1 (and see also Figure 15 and Figure 16), the initial growth of TrF can take a few epochs. This further suggests that large TrF in these scenarios is not happening in the first iteration, which might explain why there is no need (yet) to apply Fisher Penalty in the first iteration.
>
> In summary, we will revise our writing to reflect that just like warm-up in learning rate shows that using a too large learning rate from the very first iteration can be bad for generalization, using a too aggressive Fisher Penalty from the very first iteration can be bad for generalization. And analogously, just like using a large learning rate is enabled by warm-up, Figure 4 shows that using a large Fisher Penalty is beneficial when applied after a short training time (~1-3%).
>
> Thank you for raising this point. Please let us know if you have any further comments on this.
>
> **”Second, the benefit of high early regularization is somewhat inconclusive based on Fig. 10. It appears that the best generalization belongs to low early regularization, whereas high early regularization leads to better average test accuracy.”**
>
> &nbsp;&nbsp;&nbsp; Thank you for focusing on this figure. It turns out that there was a minor bug in the test data loader (we evaluated on a subset of the test set which increased the variance) used in that experiment which resulted in this unexpected behavior. We re-ran the experiment after fixing it and we have updated Figure 6 & Figure 10.
>
> We now indeed find that the runs with higher initial regularization indeed have higher best test accuracy along the trajectory compared with runs with lower initial regularization. Additionally, the higher initial regularization runs have lower values of curvature of the Hessian at the point corresponding to the best test accuracy.

---

### Official Review · AnonReviewer4 · 2020-10-29
**One step forward to uncover the implicit regularization effect of SGD.**

**Rating:** 6
**Confidence:** 4

**Review:**

After the rebuttal, the authors added a section on large-batch training, which shows that the catastrophic Fisher explosion also occurs in large batch training. This makes the paper more convincing. However, the main concern that this paper lacks theoretical contribution still exists. Therefore, I keep the weak acceptance recommendation.

Summary:

In this paper, the authors made a study on Fisher Information Matrix (FIM) in the training dynamics. By tracking the trace of FIM in the initial training process, the authors show that it can be strongly connected to a model’s generalization performance. This successfully explains why some hyper-parameter choices lead to significantly better generalization. Finally, empirical evaluations show that penalizing the trace FIM can reduce memorization and encourage wide minima solution, leading to better generalization.

Pros:

This paper takes one step forward to the question of why some hyper-parameter choices (small batch size or large learning rate) lead to better generalization. In many previous works, the implicit regularization effect of stochastic gradient descent (SGD) accounts for this observation (small batch size generalizes better). Most of the above works focused on the anisotropic noise in SGD. However, there is still a gap between the anisotropic noise and implicit regularization, i.e. why does anisotropic noise lead to implicit regularization hence better generalization. [1] explains this in a very general way: it helps to escape local minima. This paper tries to fill in the gap by leveraging FIM.

FIM is also studied in the context of SGD regularization [2]. However, it only shows that the trace of FIM affects optimization performance. How it relates to the generalization performance still remains unknown. Section 4 and section 5 shed light on this question by drawing connection to memorization and final curvatures.

Larger scale dataset such as Tiny-Imagenet is included in the experiments. This shows that the conclusion also stands on a more practical and noisy dataset.

Cons:

Most of the empirical verifications in this paper are comparing large and small learning rates. To be more general, the authors can also include the study of batch size. The central claim in this paper would be more convincing if catastrophic Fisher explosion also occurs in large batch training. More importantly, reducing the trace can improve large batch training generalization. Additionally, the connection between FIM and noisy gradient descent [3] is worth mentioning. For example, how does the trace of FIM in noisy gradient descent compared to other methods?

One of the postulations is regularizing the trace of FIM reduces memorization. The paper verifies this by introducing noisy labels. This requires modification to the training dataset. Another way to verify this postulation is to make comparisons on out-of-distribution dataset (CIFAR10-C and CIFAR100-C). The method which reduces memorization should be more robust on out-of-distribution datasets.

A majority of the analysis comes from empirical evaluations. The claim is more convincing if the conclusion can also be analytically proved in the convex quadratic scenario.
The experiments section of this paper is based on the empirical Fisher matrix (true Fisher matrix is expensive to compute). Although empirical Fisher is commonly used in practice to approximate true Fisher, the approximation is only accurate in the local minima [4]. The paper should have a discussion on the difference between true Fisher and empirical Fisher.

Legends are missing in Figure 4 and Figure 6.

[1]: Zhu, Zhanxing et al. “The Anisotropic Noise in Stochastic Gradient Descent: Its Behavior of Escaping from Sharp Minima and Regularization Effects.” ICML (2019).
[2]: Wen, Yeming et al. “An Empirical Study of Stochastic Gradient Descent with Structured Covariance Noise.” AISTATS (2020).
[3]: Wu, Jingfeng et al. “On the Noisy Gradient Descent that Generalizes as SGD.” ICML (2020)
[4]: Kunstner, Frederik et al. “Limitations of the empirical Fisher approximation for natural gradient descent.” NeurIPS (2019).

---

> ### Author Response · Authors · 2020-11-20
> **Answers to the remaining questions: CIFAR-100C is a great idea, we do estimate the trace of the actual Fisher Information Matrix,**
>
> **”One of the postulations is regularizing the trace of FIM reduces memorization. The paper verifies this by introducing noisy labels. This requires modification to the training dataset. Another way to verify this postulation is to make comparisons on the out-of-distribution dataset (CIFAR10-C and CIFAR100-C). The method which reduces memorization should be more robust on out-of-distribution datasets.”**
>
> &nbsp;&nbsp;&nbsp; This is a great idea - we will certainly add this comparison in the final version (or in this rebuttal phase if we manage to run it before the coming Tuesday).
>
> **”The experiments section of this paper is based on the empirical Fisher matrix (true Fisher matrix is expensive to compute). Although empirical Fisher is commonly used in practice to approximate true Fisher, the approximation is only accurate in the local minima [4]. The paper should have a discussion on the difference between true Fisher and empirical Fisher.”**
>
> &nbsp;&nbsp;&nbsp; We apologize for the confusion! We do compute the trace of the actual Fisher matrix. To approximate the Fisher Information Matrix, we sample y from the model prediction p(y|x), without using the training labels. Hence it is not the "empirical FIM" mentioned in [1], but rather a Monte Carlo estimate of the actual FIM. We will clarify this in the paper.
>
> **”A majority of the analysis comes from empirical evaluations. The claim is more convincing if the conclusion can also be analytically proved in the convex quadratic scenario”**
>
> &nbsp;&nbsp;&nbsp; We fully agree. While a theoretical argument would be challenging for us to add at this stage, we will make a more explicit connection to research on generalization bounds. In particular, Jiang et al. in “Fantastic Generalization Measures and Where to Find Them” (https://arxiv.org/abs/1912.02178) argue that sharpness based bounds are best correlates of generalization across multiple experiments. To the extent to which sharpness-based bounds are sensible bounds on generalization, Tr(F) can be argued to lead to tighter generalization bounds and hence better generalization.
>
> **"Legends are missing in Figure 4 and Figure 6."**
>
> Thank you. We will fix it.
>
> **"Additionally, the connection between FIM and noisy gradient descent [3] is worth mentioning."**
>
> Thank you. We will add a discussion of [1-3] to the paper.

---

> ### Author Response · Authors · 2020-11-20
> **Thank you for the review! “The central claim in this paper would be more convincing if catastrophic Fisher explosion also occurs in large batch training” - evidence suggests it does**
>
> Thank you for the review! Please find below our answer.
>
> Please let us know if you have any questions or would like us to run any additional experiments.
>
> **“The central claim in this paper would be more convincing if catastrophic Fisher explosion also occurs in large batch training”**
>
> &nbsp;&nbsp;&nbsp; We fully agree. While we didn’t manage to run all experiments with different batch sizes, we validated that catastrophic Fisher explosion also occurs in training with large mini-batch sizes.
>
> We added a section titled “Catastrophic Fisher Explosion holds in training with large batch sizes” to the appendix. For the final version we intend to rerun all the experiments in Table 1 in a setting with a large mini-batch size.
>
> As we show in Figure 16, on the CIFAR-10 dataset using a large mini-batch size does lead to Catastrophic Fisher explosion in the sense that the value of TrF (which is computed independently of the mini-batch size and is the Monte Carlo estimate of TrF) grows to a large value in the early phase of training. Training with a small batch size never exhibits such a large TrF.
>
> Crucially for the validation of the core hypothesis, in Figure 17 we show that using Fisher Penalty does reverse the negative effect on generalization of using a large mini-batch size.

---

### Official Review · AnonReviewer3 · 2020-10-29
**A through empirical study of the relationship between trace of Fisher matrix and deep learning generalization**

**Rating:** 6
**Confidence:** 4

**Review:**

Summary: This paper explores the relationship between the trace of the Fisher Information Matrix (FIM) and generalization performance of deep learning models. There are a few core insights which are verified empirically across a number of different models and datasets: (1) large learning rate/small batch sizes (which tends to lead to improved generalization) can be realized as implicitly penalizing the trace of Fisher and (2) how regularizing the trace of Fisher discourages memorization and guides optimization to "flatter minima".

My overall assessment of this paper leans towards positive. While it is probably not surprising to deep learning optimization/generalization researchers that the trace of FIM is closely related to generalization performance, this paper does a good job from the empirical standpoint, the experiments are done in a clean and systematic way to verify the hypotheses.

Some comments/remarks/questions:

- The manuscript is overall well-written; I have no problems following the logic and the experimental setups. Related works for the most part are discussed throughly and well-explained. A downside is that there is completely no theoretical arguments given in the paper, I am wondering if it's possible even in the case of a convex quadratic/extremely simple networks, the authors can show how a small Tr(F) leads to a tighter/better generalization bound?

- Just to verify- the Fisher matrix which is computed throughout is the empirical estimate of the true Fisher and not the "empirical Fisher"- as the labels are sampled from the predictive distribution? This often results in a confusion; see [1] for more details.

- In Section 3, the authors compared FP with GP which is a fair comparison. I am wondering if the authors compared their approach to even more closely-related methods such as input-output Jacobian regularization [2], regularizing by the Frobenius norm of Fisher; Tr(F^2) instead of Tr(F), and the Fisher-Rao norm which has a tight relationship with capacity/generalization of neural networks as shown in [3].

References:

[1] Kunstner, Frederik, Philipp Hennig, and Lukas Balles. "Limitations of the empirical Fisher approximation for natural gradient descent." Advances in Neural Information Processing Systems. 2019.

[2] Novak, Roman, et al. "Sensitivity and generalization in neural networks: an empirical study." arXiv preprint arXiv:1802.08760 (2018).

[3] Liang, Tengyuan, et al. "Fisher-rao metric, geometry, and complexity of neural networks." The 22nd International Conference on Artificial Intelligence and Statistics. 2019.

---

> ### Author Response · Authors · 2020-11-20
> **Thank you for the review!**
>
> Thank you for your time and the review! We are particularly happy to hear that you found the presentation and testing of the hypothesis clear.
>
> Please let us know if you have any other questions or would like us to run any additional experiments.
>
> **I am wondering if it's possible even in the case of a convex quadratic/extremely simple networks, the authors can show how a small Tr(F) leads to a tighter/better generalization bound?**
>
> &nbsp;&nbsp;&nbsp;  That is a great idea - we will add a discussion about this and connect Tr(F) better to existing literature on generalization bounds.
>
> In particular, Jiang et al. in in “Fantastic Generalization Measures and Where to Find Them” (https://arxiv.org/abs/1912.02178) argues that sharpness-based bounds are best correlates of generalization across multiple experiments. To the extent to which sharpness based bounds are sensible bounds on generalization, Tr(F) can be argued to lead to tighter generalization bounds.
>
> **”Just to verify- the Fisher matrix which is computed throughout is the empirical estimate of the true Fisher and not the "empirical Fisher"- as the labels are sampled from the predictive distribution?”**
>
> &nbsp;&nbsp;&nbsp;  Yes. To approximate the FIM, we sample y from the model prediction p(y|x), and not use the training label. Hence it is not the "empirical FIM" mentioned in [1], but is rather a Monte Carlo estimate of the actual FIM. This is mentioned below Equation 2, and more details on the penalty are provided in Appendix C.
>
> [1] Kunstner, Frederik, Philipp Hennig, and Lukas Balles. "Limitations of the empirical Fisher approximation for natural gradient descent." Advances in Neural Information Processing Systems. 2019.
>
> **”I am wondering if the authors compared their approach to even more closely-related methods such as input-output Jacobian regularization [2], regularizing by the Frobenius norm of Fisher; Tr(F^2) instead of Tr(F), and the Fisher-Rao norm.”**
>
> &nbsp;&nbsp;&nbsp;  We indeed included the results for input-output Jacobian regularization along with Fisher penalty (FP) in the paper. We call this regularization $GP_x$ in the paper and the results are shown in Figure 3 and Table 3. In Figure 3, we trained on the clean CIFAR-100 dataset with these regularizations using ResNet and VGG architectures.
>
> We found that for ResNet both regularizations perform similarly, while for VGG, FP is substantially better than FP. In Table 3, we experimented with the effect of these regularizations on preventing memorization when training on a corrupted version of the CIFAR-100 training set with different levels of label noise. We found that FP performs substantially better than GP_x in all cases. The same experiments for CIFAR-10 are shown in Figure 8 and Table 4  in the appendix. Overall, we found that FP performs best under the various settings while GP_x performs similarly under some settings but worse under others. Due to the increasing amount of resources needed, we did not try the other variants of Fisher norm, but we will definitely include references to them.

---

### Official Review · AnonReviewer5 · 2020-11-05
**Misleading and Even Wrong Writing**

**Rating:** 5
**Confidence:** 5

**Review:**

# Summary
This work claims SGD regularizes the model through penalizing the trace of Fisher in the early phase. This claim is supported by the similar generalization behavior of SGD (with optimal learning rate) and Fisher penalization (for SGD with small learning rate). A series experiments are conducted to verify the understanding.

I find the paper writing is rather misleading. On the one hand, no mathematical justifications, and the logical chain is weak --- hard to claim which factor is the cause and which one is the effect. On the other hand, though written as "trace of Fisher" in the paper, in experiments they compute a different regularizer --- norm of expected gradient. Detailed questions come in the following.

# Questions

1. Abstract, "We highlight that in the absence of implicit or explicit regularization...". How do you get rid of the implicit regularization???

2. Eq.(2) is super misleading. Trace of Fisher is the expected gradient norm, but in Eq. (2) you compute norm of expected gradient. Statistically the latter has nothing to do with the former: one is the second moment, and the other is the squared first moment. Please justify.

3. I have a very simple explanation to your observed phenomenon. In the beginning, gradient is large, thus your version of "trace of fisher" is large, as the gradient decreases, the "trace of fisher" decreases. Now we look at large learning rate and small learning rate. With large learning rate, SGD converges faster, thus its gradients decrease faster, which causes the "trace of fisher" decreases faster. But with small learning rate, SGD converges slower, thus its gradient decrease slower, and the "trace of fisher" appears to be large in the beginning epochs.

Therefore I am not at all convinced by your arguments.

4. From the above discussion, at least you need to normalize the "trace of fisher" by gradient norm --- which then becomes a measurement of gradient confusion. See following for references.

5. FP/GP. If I understand correctly, you need to compute gradient of gradient, which expands your computation graph at least twice. Can you report the GPU memory consumption?

6. Finally, let us take about the practical role of FP. According to Table 1, the improvement of FP over GP is marginal. I cannot see a potential of FP. Not only in theory, but also in practice this paper is not satisfactory.


# Missing Refs
Tons of theory paper should be discussed.  A few of them come to my brain are listed in below. Please do a more complete literature investigation.

Fisher
- Liang, Tengyuan, et al. "Fisher-rao metric, geometry, and complexity of neural networks." The 22nd International Conference on Artificial Intelligence and Statistics. 2019.
- Karakida, Ryo, Shotaro Akaho, and Shun-ichi Amari. "Universal statistics of Fisher information in deep neural networks: Mean field approach." The 22nd International Conference on Artificial Intelligence and Statistics. PMLR, 2019.

SGD regularization mechanism
- Daneshmand, H., Kohler, J., Lucchi, A., and Hofmann, T. Escaping saddles with stochastic gradients. arXiv preprint arXiv:1803.05999, 2018.
- Zhu, Zhanxing, et al. "The Anisotropic Noise in Stochastic Gradient Descent: Its Behavior of Escaping from Sharp Minima and Regularization Effects." arXiv preprint arXiv:1803.00195 (2018).
- Wu, Jingfeng, et al. "On the Noisy Gradient Descent that Generalizes as SGD." arXiv preprint arXiv:1906.07405 (2019).

Gradient confusion is highly related to the trace Fisher. See this one and its follow-ups.
- Sankararaman, Karthik A., et al. "The impact of neural network overparameterization on gradient confusion and stochastic gradient descent." arXiv preprint arXiv:1904.06963 (2019).

Adversarial regularization is also related to GP/FP:
- Miyato, Takeru, et al. "Virtual adversarial training: a regularization method for supervised and semi-supervised learning." IEEE transactions on pattern analysis and machine intelligence 41.8 (2018): 1979-1993.

---

> ### Author Response · Authors · 2020-11-17
> **Answer to “How do you get rid of the implicit regularization???” and other questions**
>
> **“How do you get rid of the implicit regularization???”**
>
> &nbsp;&nbsp;&nbsp; In our opinion, one of (many) key outstanding questions in deep learning theory is why and how high learning rate or small batch size in SGD impacts generalization. By implicit regularization we were referring to the generalization effect of a reasonably large learning rate. We suppress its regularization effect by using lower values of learning rate (which we refer to as a sub-optimal learning rate in the paper). We apologize if it were not clear, and we will improve background and motivation in the paper.
>
> **"Finally, let us take about the practical role of FP. According to Table 1, the improvement of FP over GP is marginal. I cannot see a potential of FP. Not only in theory, but also in practice this paper is not satisfactory."**
>
> &nbsp;&nbsp;&nbsp; The difference is approximately 2% for DenseNet/C100 and VGG11/C100 and 4% on SimpleCNN/C10. In these cases we would like to argue it is not marginal by the standards of the computer vision community.
>
> In the other two (out of 5) cases you are right - they are similar. However, even if they are similar, that’s actually still exciting. We are primarily excited about the theoretical importance of the finding that we can explain away an important aspect of implicit regularization in SGD. Please note that two other works were concurrently submitted to ICLR that propose only gradient norm regularization as the implicit regularizer [1,2].
>
> On a practical role of FP, let us also draw your attention to experiments involving noisy labels. In these experiments (Table 3) FP was substantially better than GP_x, and comparable to mixup. Please note that gradient norm regularization hasn’t been shown to be effective in this setting before.
>
> [1] https://openreview.net/forum?id=rq_Qr0c1Hyo
>
> [2] https://openreview.net/forum?id=3q5IqUrkcF
>
> **"From the above discussion, at least you need to normalize the "trace of fisher" by gradient norm --- which then becomes a measurement of gradient confusion. See following for references"**
>
> &nbsp;&nbsp;&nbsp;Given that we measure the trace of the actual Fisher, not “trace of Fisher”, we assume that this answers this question, but please let us know if not.
>
> **Can you report the GPU memory consumption?**
>
> &nbsp;&nbsp;&nbsp;That’s a good point. It is approximately 2-3x larger. We will add an exact measurement to the final version of the paper.
>
> **Tons of theory paper should be discussed**
>
> &nbsp;&nbsp;&nbsp;We agree with the references provided and that we missed some of important work about Fisher. We will update the related work to include them.

---

> > ### Comment · AnonReviewer5 · 2020-11-21
> > **There are many points that I cannot agree**
> >
> > 1. To my knowledge, "implicit regularization" refers to the algorithmic regularization effect, in against the explicit regularizers. As long as you use GD or SGD or large learning rate SGD, etc., there are inherent implicit regularization. I cannot see anyway to get rid of it. I strongly suggest the authors to revise the language here. It is not only about the background introduction.
> >
> > 2. The experiments are for very small datasets / very old baselines. Note CIFAR100 is a very small datasets, large models easily get overfitted thus any legitimate trick can have certain improvement. I strongly feel that in the SOTA setups, this trick cannot bring any benefits. Please at least conduct experiments against better baselines and for larger datasets like ImageNet. For example, it is easily to achieve around 85% acc for CIFAR100, then what is the meaning to compete with some 75% baselines in 2020?
> >
> > 3. The paper you cited does not make sense to me. For example, in [1], the coefficient for the regularization term scales with learning rate, which is very small in their theory, thus it basically does nothing. Please refer to Theorem 2.1 in [Kushner & Yin, 2003], which proves that as learning rate goes to zero, SGD goes to gradient flow, there is no non-trivial regularization.
> >
> >  [Kushner & Yin, 2003] Harold Kushner and G George Yin. Stochastic approximation and recursive algorithms and appli- cations, volume 35. Springer Science & Business Media, 2003.

---

> > > ### Author Response · Authors · 2020-11-23
> > > **Answers**
> > >
> > > **”To my knowledge, "implicit regularization" refers to the algorithmic regularization effect, in against the explicit regularizers**”
> > > &nbsp;&nbsp;&nbsp;
> > >
> > > &nbsp;&nbsp;&nbsp; We agree with you that implicit regularization refers to algorithmic regularization effects.  This is why our paper discusses the regularization effects due to using SGD with a large learning rate. We believe that this way to use the phrase ‘implicit regularization’ is fairly standard in the literature. It is also used in the aforementioned [1,2], as well as [3], for example.
> > >
> > > **“Please at least conduct experiments against better baselines and for larger datasets like ImageNet”**
> > > &nbsp;&nbsp;&nbsp;
> > >
> > > &nbsp;&nbsp;&nbsp; Please note that we run experiments on ImageNet (in the first part) and Tiny ImageNet (in the second part).
> > >
> > > **”I strongly feel that in the SOTA setups [...] For example, it is easily to achieve around 85% acc for CIFAR100, then what is the meaning to compete with some 75% baselines in 2020?”**
> > >
> > > &nbsp;&nbsp;&nbsp; The cited performance is achieved by (a small) DenseNet, which we believe can be still seen as a modern architecture. We will happy to try to include larger models in the final version.
> > >
> > > **”The paper you cited does not make sense to me. For example, in [1], the coefficient for the regularization term scales with learning rate, which is very small in their theory, thus it basically does nothing. Please refer to Theorem 2.1 in [Kushner & Yin, 2003], which proves that as learning rate goes to zero, SGD goes to gradient flow, there is no non-trivial regularization.”**
> > >
> > > &nbsp;&nbsp;&nbsp; Our paper, as well as the cited [1,2], argue that the implicit regularization role of using a large learning rate can be understood and introduced explicitly as gradient norm regularization. Now this effect is happening mostly in the early phase of training, see for example [4, 5]. Hence, the fact that SGD can be seen as gradient flow is not in contradiction to results in [1-5]. Happy to discuss further.
> > >
> > > References:
> > >
> > > [3] https://arxiv.org/abs/2011.02538
> > >
> > > [4] https://arxiv.org/abs/2002.09572
> > >
> > > [5] https://arxiv.org/abs/2003.02218

---

> ### Author Response · Authors · 2020-11-17
> **Why do you think Eq(2) "super misleading"? A clarification that we measure the actual Fisher, sorry for confusion**
>
> Why do you find it *“super misleading”*? The review was a bit short on this point.
>
> To corroborate the importance of regularizing the trace of the actual Fisher information matrix itself, we ran an experiment where we regularize the actual Tr(F). In short, we found that this regularization has a nearly identical effect on generalization, as well as on Tr(F), as Fisher Penalty. We added more details in the updated version of the Appendix C: “Approximations in Fisher Penalty”.
>
> Let us also clarify that in the submission we discussed the approximation immediately below Eq(2), and we also included a 2 page Appendix on this point (“Approximations in Fisher Penalty”).
>
> Finally, we are guessing that you found it misleading because you assumed that we do not compute the actual trace of the Fisher (you wrote *“your version of "trace of fisher”*) in Figure 1 and others. Consequently, perhaps you assumed there is no evidence we actually succeeded in regularizing the trace of the actual Fisher Information Matrix. However, in all the plots and tables, we report the trace of the actual Fisher Information Matrix. We will clarify this point in the submission. We hope this clarifies any confusion.
>
> Given these points above, we hope you do not find Eq.2 misleading anymore.

---

> ### Author Response · Authors · 2020-11-17
> **We understand the sentiment behind your “very simple explanation”. However, it does not agree with (published and our) evidence. Training with a large learning rate never (even for a single step) enters a region with as large TrF as the maximum TrF seen during training with a small learning rate.**
>
> Thank you for your feedback. We have updated the paper, and marked changes in red. On the whole we definitely see your rationale, and we will continue to update the paper to provide more motivation and background.
>
> Below is our rebuttal. Please let us know any other experiments you would like us to run, or if you have any questions.
>
> **”I have a very simple explanation to your observed phenomenon”**
>
> &nbsp;&nbsp;&nbsp; This is indeed not yet a widely known phenomenon and it is genuinely surprising that learning rate impacts the geometry of the trajectory over the loss surface in the early phase of training. Hence, we understand where you are coming from and agree we could have done a better job highlighting that it is now a well researched and corroborated phenomenon.
>
> The surprising effect of the learning rate on the geometry of the loss surface (e.g. the value of $Tr(\mathbf{F})$) was demonstrated in prior works cited in the introduction [1,2] ([1] was published in ICLR’20). In particular, [1] shows that training with a large learning rate escapes rapidly regions of high curvature, where the curvature was defined as the spectral norm of the Hessian evaluated at the current point on the loss surface.
>
> Perhaps the most direct evidence against the hypothesis proposed by the reviewer can be found in Figure 1 in [3] (not our work, under review at ICLR), where training with Gradient Descent rapidly finds regions of the loss surface with large curvature in the early phase (close to $\frac{2}{\eta}$) for small learning rates.
>
> We also run the following experiment to provide further evidence against the hypothesis. We train SimpleCNN on the CIFAR-10 dataset using two different learning rates, while computing the value of $Tr(\mathbf{F})$ at every *iteration*. We use $128$ random samples in each iteration. *We find that training with a large learning rate never (even for a single step) enters a region with the maximum value of $Tr(\mathbf{F})$ reaching as much as those reached with a small learning rate during training. Figure 15 in the updated paper shows the experimental data.*
>
> We also found a similar trend to hold when varying the batch size, which further suggests that the results in the paper cannot be explained by the difference in learning speed between the experiments using small and large learning rates. This result is presented in Figure 16 in the updated version.
>
> [1] https://openreview.net/forum?id=r1g87C4KwB
>
> [2] https://arxiv.org/abs/2003.02218
>
> [3] https://openreview.net/forum?id=jh-rTtvkGeM

---

### Author Response · Authors · 2020-11-21
**Summary**

We are grateful for the feedback the reviewers provided. In general, reviewers R2, R3 and R4 agree that the work is an important step towards answering one of the key questions in deep learning theory: what is the exact mechanism by which using a large learning rate impacts generalization in SGD.

The reviewers found the paper well-written, and the hypothesis well-motivated, posed, and tested. We are also happy to hear that the developed regularizer to test the hypothesis, Fisher Penalty, was interesting on its own.

We summarize below the feedback from reviewers, the extra experiments that we run, and the main points in the rebuttal.

**Pros**

* R2, R3, R4 reviewers highlight that we took an important step towards answering one of the mysteries in deep learning theory: what is the mechanism by which using a large learning rate improves generalization.
R2 wrote *“The paper's motivation is definitely relevant.  [...] The introduced Fisher-penalty is interesting and the authors provide insights into the mechanism how regularizing the trace of FIM might improve generalization. ”*.
R3 wrote “large learning rate/small batch sizes (which tends to lead to improved generalization) can be realized as implicitly penalizing the trace of Fisher”.
R4 wrote *“By tracking the trace of FIM in the initial training process, the authors show that it can be strongly connected to a model’s generalization performance. This successfully explains why some hyper-parameter choices lead to significantly better generalization”*.

* To test the hypothesis we had to develop a novel regularizer, Fisher Penalty, which was appreciated by all reviewers. In particular, R2 wrote, “The introduced Fisher-penalty is interesting and the authors provide insights into the mechanism how regularizing the trace of FIM might improve generalization.”

* R2 and R3 highlight that the hypothesis is well-posed (*“The paper is well-written and easy to follow. Authors convey their message clearly.”*) and the execution and design of experiments aimed at testing the hypothesis is clear (*“this paper does a good job from the empirical standpoint, the experiments are done in a clean and systematic way to verify the hypotheses.”.*)

**Cons. Extra experiments and answers to key questions**

* R3, R4, and R5 were confused about whether we estimate the trace of the Fisher Information Matrix (FIM), or the trace of the “Empirical” Fisher Information Matrix, which might not reflect the FIM [1]. We are sorry for writing being unclear on this. We do actually measure FIM, which is a key strength of the paper. We will clarify it in the paper.

* We added experiments showing catastrophic Fisher explosion in large batch-size training, as suggested by R4. This suggests the core claim of the paper extends to the implicit regularization role of using a small batch size.

* R5 suggested an alternative (“very simple”) explanation of our results: that catastrophic Fisher explosion does happen in training under a large learning rate, but we “miss” it due to too coarsely grained computation of Tr(F). We added a section in the appendix where we run extra experiments that show that we do measure Tr(F) with enough frequency, and catastrophic Fisher explosion does not happen in training with large learning rate.

* R5 found the Eq (2) “super misleading”. We provided additional explanations. We hope it is clear now.

* R2 pointed out that it seems inconsistent with our hypothesis (stating that implicit regularization in SGD can be explicitly introduced using Fisher Penalty) that Fisher Penalty sometimes works better when it is applied after a relatively short training time (e.g. 1-3% of the training time). We provided an argument that the wide-spread practice of gradually increasing the learning rate shows that using a large learning rate in the first iteration can also be suboptimal. This is an important point and we will clarify writing on this.

* R3 and R4 suggested adding a more theoretical basis to the claim that implicit regularization in SGD can be seen as regularizing the trace of the Fisher Information Matrix. We propose to add a more detailed discussion of generalization bounds. In particular, authors of [2] found that sharpness based bounds are best correlated with generalization across a range of experiments.




References:

[1] Kunstner, Frederik, Philipp Hennig, and Lukas Balles. "Limitations of the empirical Fisher approximation for natural gradient descent." Advances in Neural Information Processing Systems. 2019.

[2] Yiding Jiang, Behnam Neyshabur, Hossein Mobahi, Dilip Krishnan, Samy Bengio, “Fantastic Generalization Measures and Where to Find Them”, https://arxiv.org/abs/1912.02178

---

### Decision · Program_Chairs · 2021-01-07
**Final Decision**

**Decision:**

Reject

**Comment:**

This is a tricky one, hence my low confidence rating.

The reviewers seem to agree that the paper is well written, easy to follow, and that it tests a relevant hypothesis that is of interest to the community. There was some disagreement as to whether the experiments are comprehensive, complete and/or conclusive enough, although on balance it seems reviewers were overall satisfied barring a few additional requests which the authors addressed in their feedback.

However, no reviewers support the paper strongly (borderline accepts) while R5 remains unconvinced and has raised a technical point in their review about the estimator of the Trace of the Fisher information matrix. The question R5 has raised is central to the paper's methods, arguments and conclusions. In a message to ACs and PCs the authors raised concerns about R5. I personally thought that while R5 could have worded their review more carefully and respectfully (as I pointed out in my respose) the concerns raised were otherwise motivated, the reviewer engaged in a discussion, and the arguments were laid out clearly. I side with R5 and I think that the paper should be rewritten with more clarity on this question - the problem R5 found is likely to trip up others who read or build on the paper.

The authors have raised that there are two parallel submissions closely related to this one, complicating the decision making somewhat:
[1] https://openreview.net/forum?id=rq_Qr0c1Hyo [2] https://openreview.net/forum?id=3q5IqUrkcF